# Nuclear position and local acetyl-CoA production regulate chromatin state

Philipp Willnow[1,2] & Aurelio A. Teleman[1,2 ✉]

Histone acetylation regulates gene expression, cell function and cell fate[1]. Here we study the pattern of histone acetylation in the epithelial tissue of the *Drosophila* wing disc. H3K18ac, H4K8ac and total lysine acetylation are increased in the outer rim of the disc. This acetylation pattern is controlled by nuclear position, whereby nuclei continuously move from apical to basal locations within the epithelium and exhibit high levels of H3K18ac when they are in proximity to the tissue surface. These surface nuclei have increased levels of acetyl-CoA synthase, which generates the acetyl-CoA for histone acetylation. The carbon source for histone acetylation in the rim is fatty acid β-oxidation, which is also increased in the rim. Inhibition of fatty acid β-oxidation causes H3K18ac levels to decrease in the genomic proximity of genes involved in disc development. In summary, there is a physical mark of the outer rim of the wing and other imaginal epithelia in *Drosophila* that affects gene expression.

Histone acetylation regulates gene expression by reducing the compaction of chromatin and by affecting the recruitment of reader proteins such as transcriptional coactivators to DNA[1]. Histone acetylation generally correlates with transcriptionally active chromatin and is enriched in enhancers and active promoters[2]. Proper histone acetylation is required for tissue development, and altered histone acetylation is linked to cancer[3]. Although histone acetylation levels are affected by the activity of lysine acetyltransferase and deacetylase proteins, they are also regulated by the cellular metabolic state. For instance, glycolytic rates influence cellular acetyl-CoA levels, which in turn affect histone acetylation[4–6]. Furthermore, sirtuin histone deacetylases (HDACs) require NAD+ as a cofactor, which further links cellular redox state to histone acetylation.

Another important factor in tissue development is the positioning of nuclei within cells, which is actively regulated by interactions with the cytoskeleton[7]. During mitosis, nuclei move from basal to apical regions of the epithelium to enable them to round up for cytokinesis and to allow oriented cell divisions[8]. Nuclear positioning is also thought to have an instructive role in other biological processes such as Notch signalling[9,10]. In this study, we show that the position of a nucleus within an epithelium determines its epigenetic state and therefore gene expression and cell specification.

## H3K18ac is high in the wing disc rim

To study whether histone acetylation and cellular metabolism are uniform in a growing tissue, we selected the *Drosophila* wing imaginal disc. This structure comprises an epithelial bilayer with a thin, squamous peripodial membrane and a thicker disc proper that gives rise to adult structures, including the wing (Fig. 1a). The disc proper is a pseudostratified epithelium in which every cell spans from basal to apical surfaces, but they are packed so tightly that nuclei sit at different positions giving the impression of multilayering (Fig. 1b). Levels of total histone 3 (H3), H3K9 acetylation (H3K9ac), H3K27ac and

H3 methylation were uniform in the disc (Fig. 1c and Extended Data Fig. 1a), whereas H3K18ac and H4K8ac were enriched in the rim of the disc (Fig. 1d and Extended Data Fig. 1b). This pattern was present in fully grown wandering third instar larvae and earlier in development (Extended Data Fig. 1c,d). All antibody signals had the expected sub-nuclear localization, in which they were either enriched (H3K9me2 and H3K9me3) or excluded (H3K9ac, H3K27ac, H3K18ac, H4K8ac and H3K36me3) from highly compacted DNA (bright DAPI puncta; Extended Data Fig. 1e). Total acetylated lysine (ac-K) was also mainly nuclear (Extended Data Fig. 1e) and enriched in the rim (Fig. 1e). Hence, overall lysine acetylation, exemplified by H3K18ac and H4K8ac, is enriched in the rim, whereas H3K9ac and H3K27ac represent only minor fractions of total histone acetylation. For follow-up experiments, we selected H3K18ac as a representative mark.

## H3K18ac is high in rim nuclei

To understand the mechanisms that underlie this pattern, we asked whether regions with high levels of H3K18ac (H3K18ac^high) correspond to a specific gene expression domain. To test this, we used the teashirt-GAL4 (tshG4) driver, which is expressed in proximal H3K18ac^high cells of the disc (GFP in Fig. 1f) and combined it with temperature-sensitive GAL80 (G80^ts) to induce the expression of the apoptotic gene *reaper*. At 24 h after induction, the tsh expression domain was ablated (no GFP+ cells; Fig. 1g). However, a new outer rim of H3K18ac^high nuclei with a different cell identity (tsh− and GFP−) formed, which suggested that H3K18ac^high is not determined by cell identity. Furthermore, the new outer rim of H3K18ac^high included the dorsal edge of the pouch (Fig. 1g, arrowhead), which is normally internal to the disc and H3K18ac^low (Fig. 1f, arrowhead). Thus, pouch cells became H3K18ac^high if placed at the outer rim of the disc owing to ablation of their neighbours. Disc cross-sections revealed that the H3K18ac^high region did not constitute a contiguous area of epithelium (arrowheads in Fig. 1h,j and Extended Data Fig. 1f). Even within a H3K18ac^high region,

[1]German Cancer Research Center (DKFZ), Heidelberg, Germany. [2]Heidelberg University, Heidelberg, Germany. ✉e-mail: a.teleman@dkfz.de

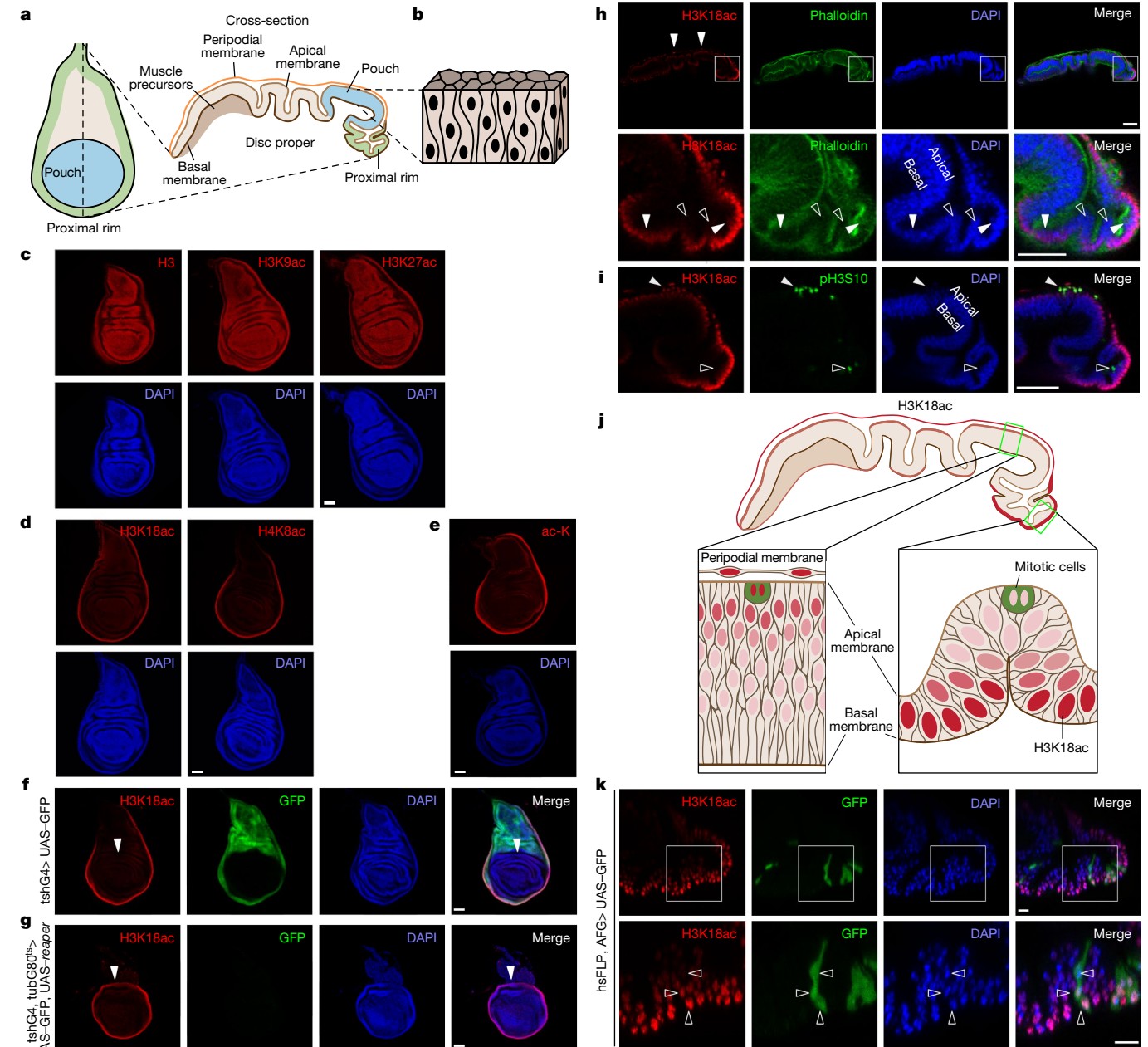

**Fig. 1 | Histone acetylation is non-uniform in wing discs. a,b**, Schematic representation of a wing disc (**a**) and a pseudo-stratified epithelium (**b**). **c**,**d**, Total H3, H3K9ac and H3K27ac are uniformly distributed in the wing disc (**c**; *n* = 17, 7 and 8 discs from left to right), whereas H3K18ac and H4K8ac are enriched in the tissue rim (**d**; *n* = 16 and 12 discs from left to right). **e**, Total ac-K is enriched in the wing disc rim (*n* = 13 discs). **f**,**g**, H3K18ac^high is not a feature of cell identity. Although the H3K18ac^high region is normally in the tsh domain (**f**; *n* = 8 discs), elimination of this domain through tshG4^ts-driven expression of *reaper* (**g**; *n* = 9 discs) causes the formation of a new H3K18ac^high rim (including the dorsal edge of the pouch, arrowheads), which does not express tshG4. Temperature-sensitive expression of reaper was induced for 1 day at 29 °C. **h**, Cross-sections of wing discs show increased H3K18ac in nuclei facing the outside of the tissue (filled arrowheads) but not in nuclei facing the interior (open arrowheads). White boxes in the top panels indicate the zoomed-in areas in the bottom panels (*n* = 7 discs). **i**, H3K18ac levels do not correlate with cell cycle phase. Mitotic pH3^+ nuclei have H3K18ac^high in the wing pouch (filled arrowheads) but low H3K18ac in proximal regions (open arrowheads) (*n* = 11 discs). **j**, Schematic representation of the H3K18ac pattern (red). Mitotic cells in green. **k**, Clonal heritage does not define H3K18ac levels. Arrowheads indicate three nuclei of a GFP-expressing clone (see Methods for details) with high (bottom arrowhead), medium (middle arrowhead) or low H3K18ac (top arrowhead) depending on their position within the wing disc tissue. The white boxes in the top panels indicate the zoomed-in areas in the bottom panels (*n* = 29 discs). Scale bars, 50 μm (**c**–**i**) or 10 μm (**k**).

cells with nuclei facing the tissue interior had H3K18ac^low (open arrowheads in Fig. 1h, bottom, open arrowheads in Fig. 1i). Thus, neighbouring cells have different levels of H3K18ac that correlate with the position of their nuclei (Fig. 1j).

H3K18ac was also mildly increased in the apical wing pouch and high in nuclei of the peripodium (Fig. 1h, arrowheads). As nuclei in the wing disc move apically for mitosis[9], we asked whether the H3K18ac^high nuclei are mitotic. In the wing pouch, mitotic nuclei were apical and H3K18ac^high (Fig. 1i, filled arrowheads). In the rim, they were also apical (Fig. 1i, open arrowheads) but distant from the tissue surface and H3K18ac^low. Hence, H3K18ac status does not correlate with the cell cycle phase (Fig. 1j).

Because nuclei are continuously cycling in the tissue, H3K18ac must be dynamically changing. In rim cells, H3K18ac decreases when mitotic nuclei move apically and increases when they move basally. To test this

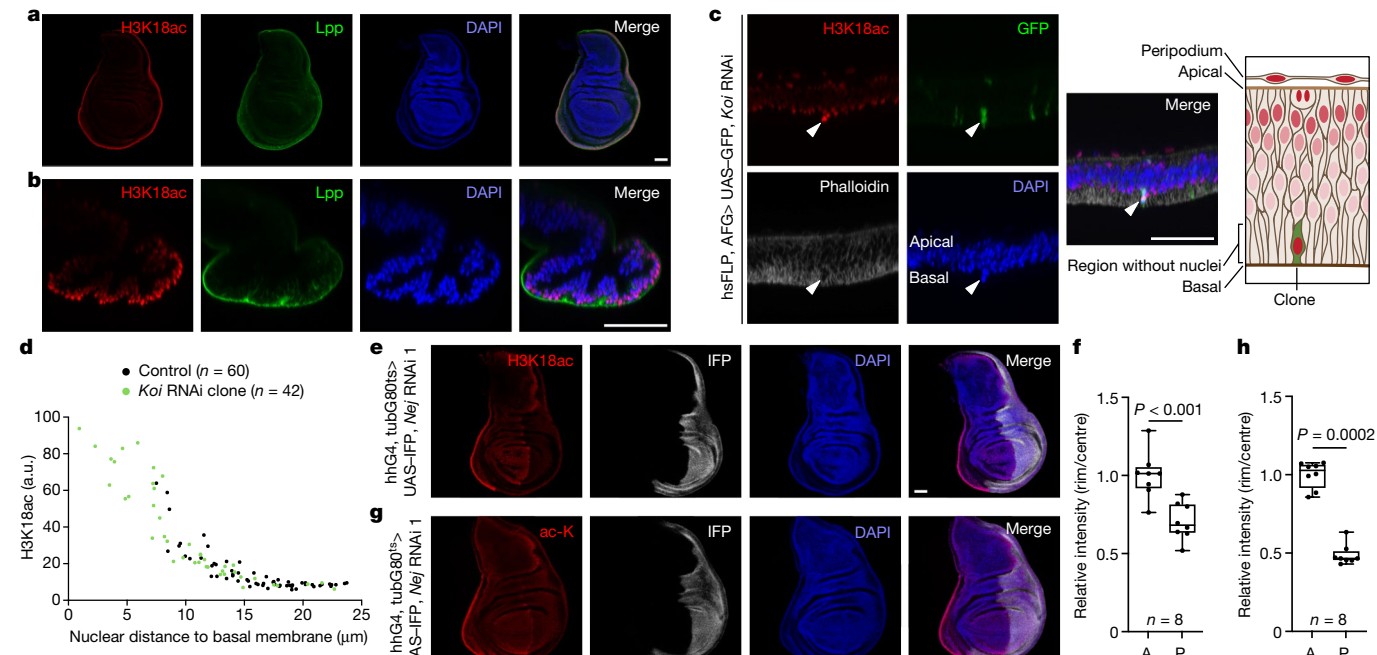

**Fig. 2 | Histone acetylation is high in outward-facing nuclei. a,b,** H3K18ac[high] nuclei are facing the outside of the wing disc tissue and are close to the haemolymph, as seen by immunostaining for haemolymph-derived Lpp, seen in *x–y* section (**a**) or cross-section (**b**). *n* = 21 discs. **c,d,** Nuclei in *Koi* loss-of-function (RNAi) clones in the wing disc move further basally than normal, where they come in proximity to the tissue surface and have high H3K18ac. Representative images in **c** (*n* = 19 discs) and quantified (in arbitrary units (a.u.)) in **d**. Nuclei from control cells and *Koi* loss-of-function clones are indicated in black and green, respectively. Schematic in **c** shows the position of the clone cell and its elevated H3K18 acetylation. Arrowheads indicate clone cells. **e,f,** H3K18 is acetylated by Nej, as *Nej* knockdown causes complete loss of H3K18ac in the posterior compartment (IFP⁺). Temperature-sensitive knockdown of *Nej* was induced for 1 day at 29 °C. Representative images in **e** and quantified in **f**. Significance by Mann–Whitney test (two-sided). **g,h,** Knockdown of *Nej* results in a reduction in total ac-K in the posterior compartment (IFP⁺), which indicates a central role for Nej in protein acetylation in the wing disc. Temperature-sensitive knockdown of *Nej* was induced for 1 day at 29 °C. Representative images in **g** and quantified in **h**. Significance by Mann–Whitney test (two-sided). Box plots show centre line (median), box limits (first and third quartiles) and whiskers (outer data points). Scale bars, 50 μm (**a–c,e,g**). A, anterior; P, posterior.

idea, we generated small clones stably expressing GFP to enable lineage tracing of the progeny of one cell. Within one clone, cells had different levels of H3K18ac (Fig. 1k, arrowheads). The 3 cells analysed were separated by 1–2 cell cycles; therefore H3K18ac levels can change within a cell cycle. This finding also confirms that H3K18ac levels do not depend on cell identity, and suggests that H3K18ac increases when nuclei reach the outside surface of the tissue. We marked the outer surface of the wing disc by staining for lipoprotein (Lpp)[11], which is secreted into haemolymph by the fat body and taken up by wing disc cells exposed to haemolymph. H3K18ac levels were high in regions stained positive for Lpp (Fig. 2a,b and Extended Data Fig. 1g–g′). When rim cells were ablated with tshG4, G80[ts]> *reaper*, the newly haemolymph-exposed (Lpp⁺) region became H3K18ac[high] (Extended Data Fig. 1h–h′). To confirm that nuclear position regulates H3K18ac, we generated clones of cells with loss-of-function for klaroid (Koi) or Rho kinase (Rok), which affect nuclear positioning through the cytoskeleton. Normally, the basal side of the wing pouch is devoid of nuclei (Fig. 2c). Hence, the most basal nuclei are distant from the tissue surface and are not H3K18ac[high]. Some Koi or Rok mutant nuclei migrated more basally than normal, approaching the tissue surface, and had increased H3K18ac (Fig. 2c,d and Extended Data Fig. 1i,j). Thus, nuclei become H3K18ac[high] when they are near the tissue surface.

## Acetylation activity is high in the rim

High rim H3K18ac can be caused either by uniform acetylation in the disc and high deacetylation in the centre or by high acetylation in the rim. To distinguish between these possibilities, we first identified the enzymes acetylating and deacetylating H3K18.

To identify the acetyltransferase responsible for acetylating H3K18 in the wing disc, we knocked down *nejire* (*Nej*), *general control non-repressed protein 5* (*Gcn5*) or *elongator complex protein 3* (*Elp3*)—all linked to H3K18ac[1]—in the posterior compartment of the wing disc (GFP⁺) and analysed H3K18ac levels (Extended Data Fig. 2a–d). Only knockdown of *Nej* caused H3K18ac loss, albeit with reduced compartment size. However, a short 1-day knockdown of *Nej*, during which tissue size is not yet affected, with two independent RNA-mediated interference (RNAi) lines also caused loss of H3K18ac and total ac-K (Fig. 2e–h and Extended Data Fig. 2e,f). This result indicated that Nej is responsible for most of the lysine acetylation in wing discs.

Notably, a brief 1-day knockdown of *Nej* in the proximal region using tshG4 removed H3K18ac from the rim (except in a region where tshG4 is not expressed) (Extended Data Fig. 2g–g″). However, it did not ablate the tissue and did not lead to the formation of a new H3K18ac[high] domain at the outer region of wild-type tissue (GFP⁻; Extended Data Fig. 2g′–g″, arrowheads). This finding is in contrast to the result obtained when proximal tissue is ablated with tshG4> *reaper*, which led to a new H3K18ac[high] region (Fig. 1g). Hence, a new rim of H3K18ac does not form when the outer rim of H3K18ac is gone, but forms when the outer rim of cells is removed.

To identify the H3K18ac deacetylase in the wing disc, we set up a H3K18ac deacetylation assay. Pharmacological inhibition of Nej with A485 in explants led to loss of H3K18ac within 1 h (Extended Data Fig. 2h,i). This loss, owing to deacetylase activity, should be blocked if the responsible deacetylase is inhibited or knocked down. Neither pharmacological inhibition nor genetic knockdown of the NAD⁺-dependent HDACs (sirtuins) prevented the H3K18ac reduction caused by A485 (Supplementary Data 1). Instead, inhibition of the

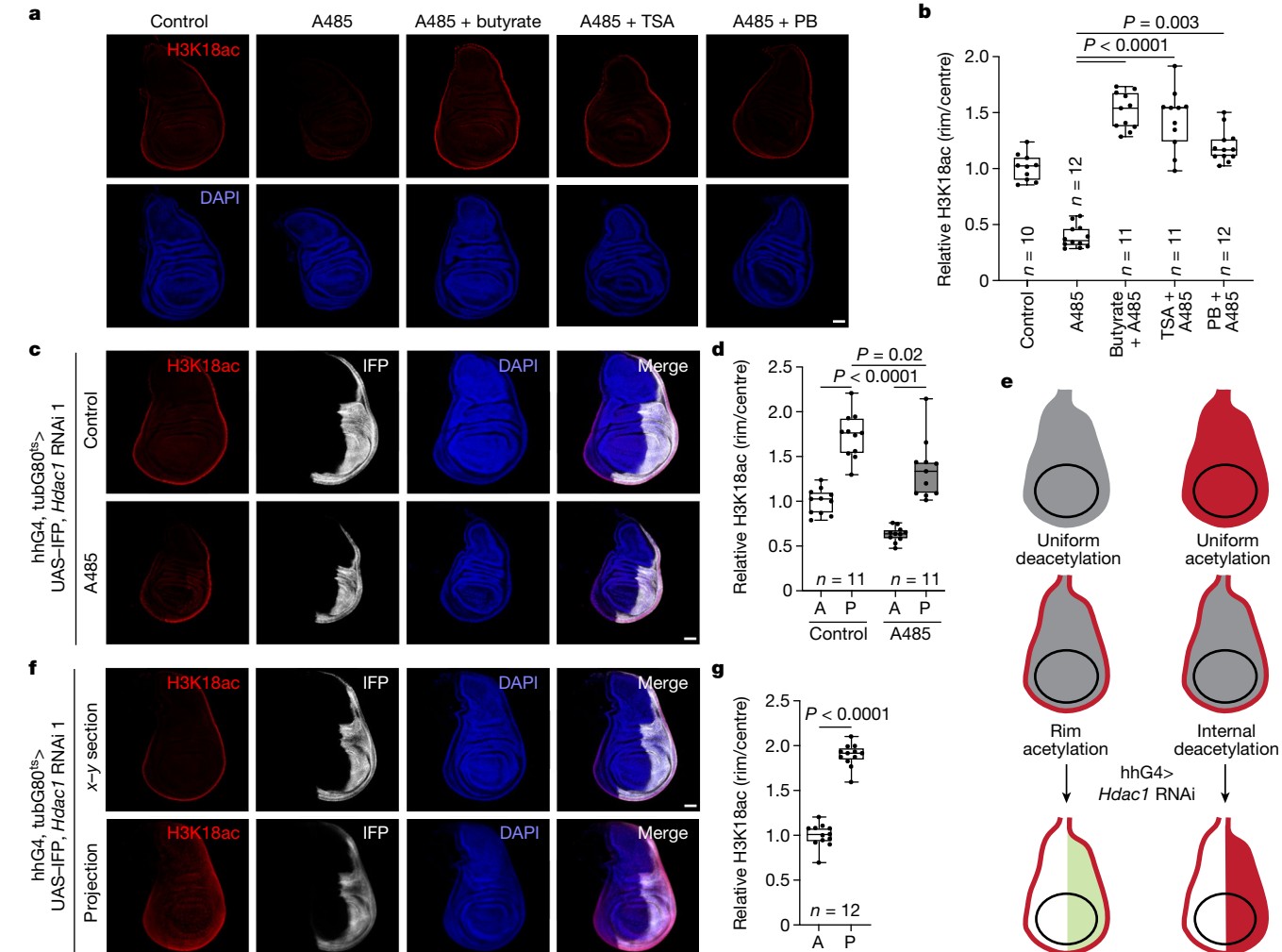

**Fig. 3 | High rim histone acetylation is caused by localized acetylation not deacetylation. a,b**, H3K18ac is deacetylated by a class I, II or IV HDAC, as inhibition with butyrate (20 mM), TSA (500 nM) or PB (100 nM) causes retention of H3K18ac[high] after Nej inhibition by A485 (20 μM). Discs were incubated with inhibitors for 2 h in explant cultures. Representative images in **a** and quantified in **b**. Significance by Kruskal–Wallis test with Dunn's multiple comparisons test. **c,d**, HDAC1 is the main H3K18ac-specific deacetylase because *Hdac1* knockdown prevents loss of H3K18ac after Nej inhibition by A485 (20 μM) in the posterior compartment (IFP⁺). Temperature-sensitive knockdown of *Hdac1* was induced for 2 days at 29 °C. Discs were incubated for 1 h in explant cultures in the presence or absence of A485. Representative images in **c** and quantified in **d**. Significance by two-way analysis of variance (ANOVA) with Šídák's multiple comparisons test. **e**, H3K18ac[high] (red) in the rim of the wing disc may be achieved by uniform deacetylation and local acetylation in the tissue rim (left) or by uniform acetylation and local deacetylation in the tissue interior (right). Depending on the mechanism, knockdown of *Hdac1* in the posterior compartment (green) should either increase H3K18ac levels only in the rim or in the entire compartment, respectively. **f,g**, *Hdac1* knockdown causes a mild increase in H3K18ac in the posterior compartment (IFP⁺) as show in *x–y* section (top) or sum projection (bottom). Temperature-sensitive knockdown of *Hdac1* was induced for 2 days at 29 °C. H3K18ac levels from *x–y* sections (**f**) quantified in **g**. Significance by Mann–Whitney test (two-sided). Box plots show centre line (median), box limits (first and third quartiles) and whiskers (outer data points). Scale bars, 50 μm (**a,c,f**).

metabolite-independent HDACs (classes I, II and IV) with butyrate, trichostatin A (TSA) or panobinostat (PB) maintained high H3K18ac levels in the presence of A485 (Fig. 3a,b). A RNAi screen revealed that knockdown of *Hdac1* led to a mild increase in H3K18ac and a stronger increase in H3K9ac and H3K27ac (Supplementary Data 2 and 3). Hence, we tested the knockdown of *Hdac1* in the deacetylation assay. *Hdac1* knockdown prevented the reduction in H3K18ac caused by A485 (Fig. 3c,d), thereby identifying HDAC1 as the main H3K18ac deacetylase in the wing disc. For this experiment, we induced *Hdac1* knockdown for only 1 day because prolonged knockdown caused reduced compartment size (Supplementary Data 2). The effects of knocking down acetyltransferases and deacetylases on H3K9ac, H3K18ac, H3K27ac, H4K8ac and disc morphology are summarized in Supplementary Table 1.

These results enabled us to determine whether the H3K18ac pattern is due to high rim acetylation or high deacetylation in the interior. If it was caused by deacetylation in the interior, then knockdown of *Hdac1*, or combined pharmacological inhibition of all deacetylases, should increase interior H3K18ac to rim levels. However, this was not the case (Fig. 3e–g and Extended Data Fig. 3a,b). Hence, the H3K18ac pattern is caused by high acetylation in the rim and not by high deacetylation in the interior (Fig. 3e). This result is consistent with the finding that HDAC1 levels are uniform in the disc and not enriched in the pouch (Extended Data Fig. 3c–e).

## Acetyl-CoA from fatty acid β-oxidation

Next, we wanted to understand why rim nuclei have high acetylation levels. Potential explanations include high Nej expression or activity or high availability of the substrate acetyl-CoA in the rim. *Nej* mRNA and protein levels, however, were not increased in the rim (Fig. 4a,b

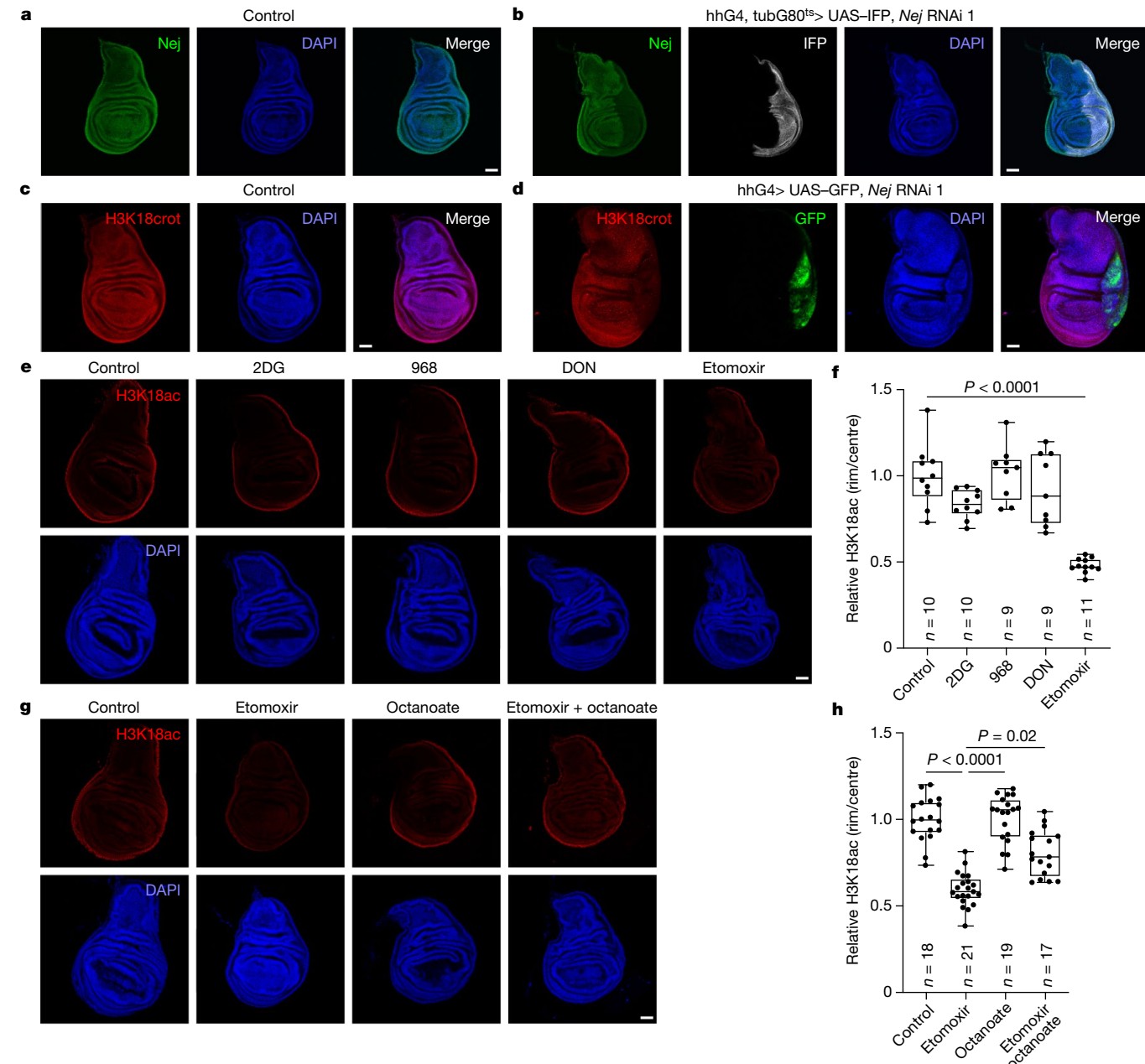

**Fig. 4 | FABO provides acetyl-CoA for acetylation of H3K18. a,b,** Nej is uniformly expressed in a control wing disc (**a**; $n = 21$ discs). Specificity of the antibody was validated by *Nej* knockdown, which causes a decrease in the immunosignal (**b**; hhG4, tubG80[ts]> UAS–IFP, *Nej* RNAi 1; $n = 11$ discs). Temperature-sensitive knockdown of *Nej* was induced for 1 day at 29 °C in the posterior compartment (IFP[+]). **c,d,** H3K18crot, as a readout for *Nej* activity, is uniform in a control wing disc (**c**; $n = 15$ discs) and mediated by Nej because *Nej* knockdown causes loss of the modification (**d**; hhG4, tubG80[ts]> UAS–IFP, *Nej* RNAi 1; $n = 11$ discs) in the posterior compartment (GFP[+]). **e,f,** Rim H3K18ac is reduced after inhibition of FABO (500 μM etomoxir) but not glycolysis (20 mM 2DG) or glutaminolysis (50 μM 968 or 500 μM DON) for 2 h in explant cultures. Representative images in **e** and quantified in **f**. Significance by Kruskal–Wallis test with Dunn's multiple comparisons test. **g,h,** Etomoxir-induced (500 μM) loss of H3K18ac is partially rescued by supplementation with the short-chain fatty acid octanoate (10 mM). Discs were incubated with the indicated compounds for 2 h in explant cultures. Representative images in **g** and quantified in **h**. Significance by Kruskal–Wallis test with Dunn's multiple comparisons test. Box plots show centre line (median), box limits (first and third quartiles) and whiskers (outer data points). Scale bars, 50 μm (**a**–**e**,**g**).

and Extended Data Fig. 4a–f). To assay Nej activity independently of acetyl-CoA levels, we used the fact that Nej also transfers crotonyl-CoA onto H3K18 to produce crotonylated H3K18 (H3K18crot)[12]. If Nej activity is increased in the rim, H3K18crot should also show a rim pattern, but this was not the case (Fig. 4c). We validated that Nej is responsible for H3K18crot in the wing disc through *Nej* knockdown (Fig. 4d). As the only difference between uniform H3K18crot and non-uniform H3K18ac by Nej is the substrate acetyl-CoA, this result suggests that nuclear acetyl-CoA levels are increased in the rim. Alternatively, the interaction of Nej with proteins that help it differentially acetylate or crotonylate might be regulated in the rim.

Several pathways produce acetyl-CoA for protein acetylation. These include glycolysis (through pyruvate dehydrogenase (Pdh)), glutaminolysis and fatty acid β-oxidation (FABO). Pharmacological inhibition of glycolysis with 2-deoxyglucose (2DG) or of glutaminolysis with glutaminase inhibitors 968 or 6-diazo-5-oxo-L-norleucin (DON) did not

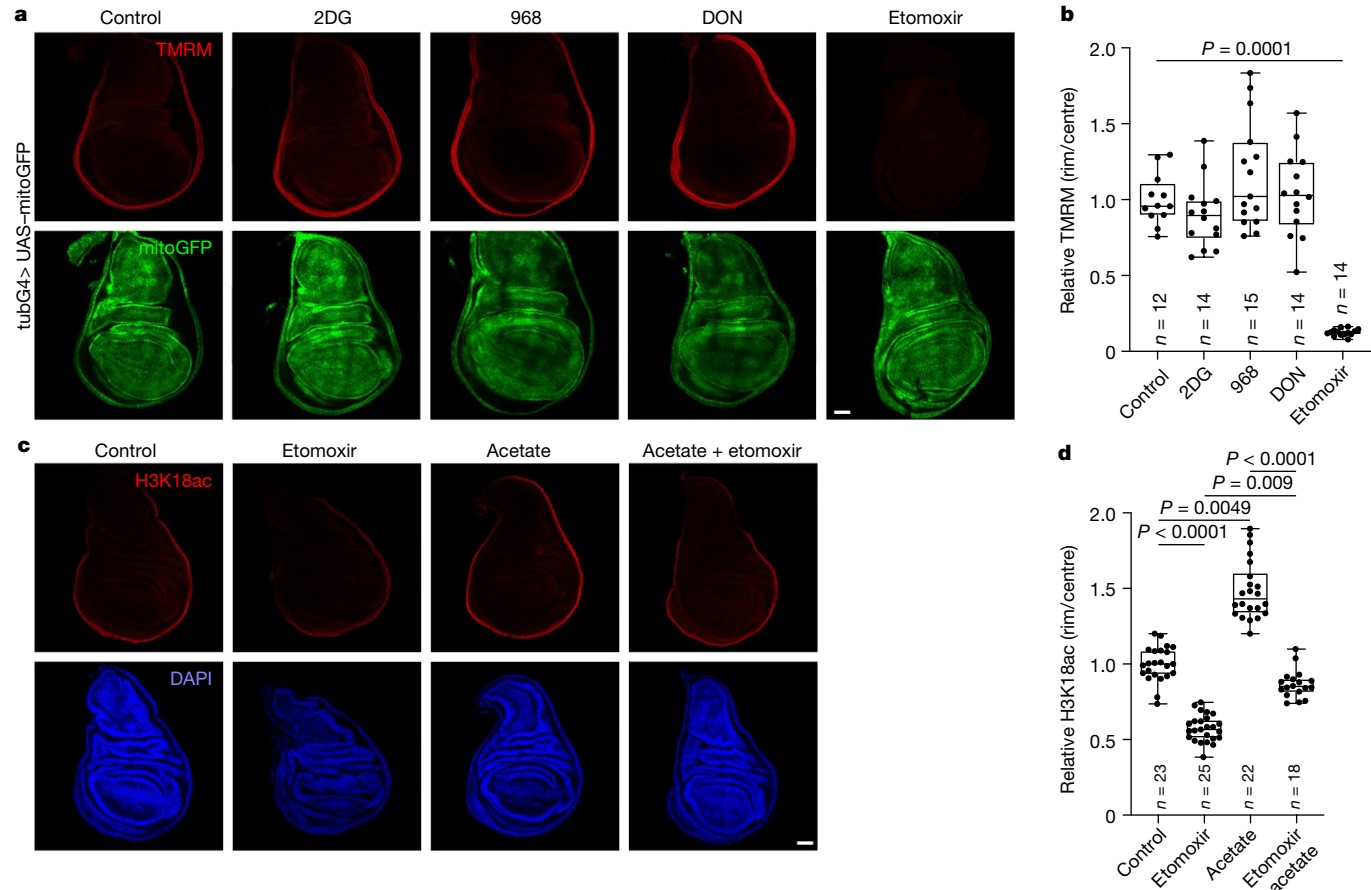

**Fig. 5 | FABO is high in the disc rim region. a,b,** MMP detected by TMRM staining is high in the rim and decreases after inhibition of FABO (500 μM etomoxir) but not glycolysis (20 mM 2DG) or glutaminolysis (50 μM 968 or 500 μM DON) for 1 h in explant cultures. Mitochondria are visualized with ubiquitously expressed mitoGFP. Representative images in **a** and quantified in **b**. Significance by Kruskal–Wallis test with Dunn's multiple comparisons test.

**c,d,** Loss of H3K18ac after inhibition of FABO (500 μM etomoxir) for 2 h in explant cultures is rescued by acetate (10 mM). Representative images in **c** and quantified in **d**. Significance by Kruskal–Wallis test with Dunn's multiple comparisons test. Box plots show the centre line (median), box limits (first and third quartiles) and whiskers (outer data points). Scale bars, 50 μm (**a,c**).

reduce H3K18ac levels in the wing disc (Fig. 4e,f). To further exclude glycolysis through Pdh, we knocked down either *Pdha* or *Pdh kinase* (*Pdk*), which phosphorylates and inhibits Pdh. Knockdown of both factors were efficient, as judged by loss of the Pdk-dependent phosphorylation on Pdha (pS293) (Extended Data Fig. 4g,i). Nonetheless, *Pdha* or *Pdk* knockdown did not decrease or increase H3K18ac levels, respectively (Extended Data Fig. 4h,j).

Instead, blocking FABO with etomoxir strongly decreased rim H3K18ac (Fig. 4e,f and Extended Data Fig. 4k,l), which correlated with the degree of inhibition of FABO by etomoxir in *Drosophila*[13]. As high concentrations of etomoxir can also inhibit mitochondrial complex I, we tested a canonical complex I inhibitor, rotenone, but it did not phenocopy etomoxir (Extended Data Fig. 4m–p). As a further control for etomoxir specificity, we treated wing discs with octanoate. Etomoxir inhibits FABO by preventing active mitochondrial import of long-chain fatty acids. Short-chain fatty acids such as octanoate, however, diffuse freely into mitochondria. Treatment of wing discs with octanoate partially rescued the reduction in rim H3K18ac caused by etomoxir (Fig. 4g,h), which indicated that etomoxir is working on-target. Thus, FABO is the main source of acetyl-CoA for H3K18ac in the rim. As expected, a brief knockdown of *Hdac1* in the posterior compartment partially rescued the reduction in H3K18ac caused by etomoxir (Extended Data Fig. 4q,r).

Notably, etomoxir also decreased rim levels of H4K8ac, H3K9ac, H3K27ac and total acetyl-lysine (Extended Data Fig. 5a–f). Thus, FABO

is the main source of acetyl-CoA for lysine acetylation in the rim in general. A possible explanation for why H3K9ac and H3K27ac levels are uniform is that H3K9 and H3K27 are heavily methylated[14] and, hence, blocked (Extended Data Fig. 5g). Indeed, H3K9ac and H3K27ac levels are low because they do not contribute much to the non-uniform level of total lysine acetylation (Fig. 1c,e).

## FABO is high in the rim

We next asked whether FABO is also increased in the rim of the wing disc. Because FABO contributes to the mitochondrial membrane potential (MMP), we first visualized the MMP using tetramethylrhodamine-methylester (TMRM). Although mitochondria are ubiquitously present throughout the wing disc (mitochondrially targeted GFP (mitoGFP) signals; Fig. 5a), the MMP was high only in a proximal region of the wing disc that was broader than the H3K18ac^high region (Fig. 5a). The TMRM signal was lost following mitochondrial depolarization with carbonyl cyanide-*p*-trifluoromethoxyphenylhydrazone (FCCP), thereby validating the staining method (Extended Data Fig. 5h,i). Various metabolic pathways fuel respiration, including glycolysis, glutaminolysis and FABO. We inhibited each pathway to see which one predominantly contributes to the high rim MMP. Pharmacological inhibition of glycolysis with 2DG or of glutaminolysis with 968 or DON, as well as knockdown of *Pdha* or *Pdk* did not affect the MMP (Fig. 5a,b and Extended Data Fig. 5j,k). Instead, inhibition of FABO substantially

reduced the MMP (Fig. 5a,b). Thus, in the proximal region of the wing disc, the MMP is mainly fuelled by FABO. We confirmed these findings using the genetic, mitochondrially localized, ratiometric pH-sensor SypHer3s-dmito[15] as an independent readout for the MMP. As the MMP is generated by pumping protons out of the mitochondrial matrix, mitochondrial pH increases with higher MMP. Ratiometric imaging of SypHer3s-dmito-expressing discs confirmed that the MMP is high in the rim (Extended Data Fig. 5l). Moreover, it was abolished by FCCP (Extended Data Fig. 5l) and was not affected by glycolysis or glutaminolysis inhibitors, but was substantially decreased by etomoxir (Extended Data Fig. 5m). Because the MMP mainly reflects FABO in the wing disc, these results indicate that FABO is high in a proximal rim of cells. Consistent with this result, lipid droplets, visualized using BODIPY, were enriched in the pouch and depleted in proximal regions (Extended Data Fig. 5n), which anti-correlated with FABO.

## H3K18ac is not upstream of FABO

These results indicated that FABO is high in a proximal domain of the wing disc, where it generates acetyl-CoA for H3K18ac. As inhibition of FABO causes loss of H3K18ac, FABO is upstream of H3K18ac. Conversely, knockdown of *Nej* in the posterior compartment using two RNAi lines did not affect the MMP (Extended Data Fig. 5o,p), thereby establishing a clear epistatic relationship with FABO upstream of H3K18ac.

## Nuclear versus mitochondrial position

We tested whether H3K18ac^high nuclei are the ones located near mitochondria. However, this was not the case because in the rim, mitochondria were subcellularly localized both apically and basally, close to nuclei that are both H3K18ac^high and H3K18ac^low (Extended Data Fig. 5q–q′). Single-cell clones expressing mitoGFP and membrane-anchored infrared fluorescent protein (IFP) confirmed that single cells have mitochondria both in apical (filled arrowheads) and basal regions (open arrowheads), regardless of nuclear position and H3K18ac state (Extended Data Fig. 5r–r′). Indeed, H3K18ac^high and H3K18ac^low nuclei were equally close to mitochondria (Extended Data Fig. 5s). We next asked whether nuclei that are H3K18ac^high are the ones located close to mitochondria with high membrane potential. However, in rim regions, both the mitochondria facing the tissue outside and the ones facing inside had high MMP (Extended Data Fig. 6a), as the region with high mitochondrial membrane is broader than the H3K18ac^high region. In summary, we could not find evidence to support the hypothesis that the relative position of nuclei and mitochondria determines H3K18ac levels.

## ACSS2 generates acetyl-CoA for H3K18ac

Acetyl-CoA produced by FABO is localized inside mitochondria and needs to be exported for nuclear histone acetylation (Extended Data Fig. 6b). Export occurs either as citrate, which is converted back to acetyl-CoA in the cytosol by ATP citrate lyase (*Drosophila*: Atpcl; human: ACLY), or as acetate, which is converted back by acetyl-CoA synthase (*Drosophila*: AcCoAS; human: ACSS2). To determine which pathway is required for H3K18ac in the wing disc, we knocked down *ACLY* in the posterior compartment (Extended Data Fig. 6c), but this did not reduce H3K18ac levels (Extended Data Fig. 6d). Consistent with this result, citrate cannot act as an exogenous acetyl-CoA source to rescue the loss of H3K18ac caused by etomoxir (Extended Data Fig. 6e,f). By contrast, pharmacological inhibition of ACSS2 with an ACSS2 inhibitor (ACSS2i) strongly decreased H3K18ac levels (Extended Data Fig. 6g,h). Furthermore, acetate rescued the reduction in H3K18ac caused by etomoxir (Fig. 5c,d), and this rescue is due to ACSS2 because it was blocked by ACSS2i (Extended Data Fig. 6i,j). In summary, in the wing disc, acetyl-CoA generated by FABO is exported from mitochondria as acetate and converted back to acetyl-CoA by ACSS2 for histone acetylation.

## ACSS2 activity in the rim

We noted that exogenous acetate specifically increases H3K18ac in the rim of etomoxir-treated discs, rather than ubiquitously throughout the tissue (Fig. 5c). This can be either technical, if exogenous acetate does not penetrate the tissue efficiently, or biological if acetate is only converted to nuclear acetyl-CoA in the rim. Nuclear and cytosolic production of acetyl-CoA are regulated separately[16]. Hence to test whether exogenous acetate reaches all cells of the wing disc, we assayed acetylation of α-tubulin on lysine 40 (tubK40ac), which depends on cytosolic and not nuclear acetyl-CoA. Treatment of discs with acetate caused an increase in tubK40ac throughout the disc (Fig. 6a,b), which indicated that exogenous acetate reaches all regions of the tissue. Therefore, the difference between cytosolic tubK40ac and nuclear H3K18ac implies that conversion of acetate into nuclear acetyl-CoA is increased in nuclei facing the tissue exterior. Consistent with this idea, acetate treatment increased rim levels of the Nej targets H3K18ac, H4K8ac, total ac-K and H3K27ac (Extended Data Fig. 7a–d′,f,g) and of the Gcn5 target H3K9ac (Extended Data Fig. 7e–e′,f,h). These results confirm that the rim effect is not specifically due to increased Nej catalytic activity. Nuclear conversion of acetate to acetyl-CoA was also identified as the rate-limiting step controlling H3K18ac in the rim.

ACSS2 translocates to the nucleus to generate nuclear acetyl-CoA for histone acetylation[17,18]. To examine whether this mechanism contributes to the increased rim acetylation of H3K18, we analysed ACSS2 subcellular localization in the wing disc. As we could not obtain a RNAi line that efficiently knocks down *ACSS2* to confirm specificity of the antibody signal, we instead overexpressed ACSS2 in the posterior compartment so that the additional signal in this region must be due to ACSS2 (Extended Data Fig. 8a). Wing disc cross-sections revealed higher nuclear levels of ACSS2 in rim nuclei than in nuclei facing the tissue interior (Fig. 6c,d). We observed the same phenotype when inspecting endogenous ACSS2 levels (Fig. 6e–f and Extended Data Fig. 8b), which indicated this is not an overexpression artefact. As a negative control, we quantified levels of another nuclear protein, H3, and found that they do not depend on nuclear distance to the tissue surface (Extended Data Fig. 8c–e). To test whether nuclear position regulates the amount of nuclear ACSS2, we quantified nuclear ACSS2 in Koi and Rok mutant clones in the pouch. Indeed, re-localization of nuclei closer to the tissue surface through these genetic perturbations increased nuclear ACSS2 levels (Extended Data Fig. 8f–i). In summary, spatial cues regulate both an increase in FABO in rim regions of the disc and an increase in nuclear ACSS2 levels in nuclei facing the tissue exterior (Fig. 6g). Although both factors probably contribute to the high H3K18ac in these nuclei, we cannot exclude that ACSS2 activity is also activated through additional mechanisms.

## Effect on gene expression

Histone acetylation regulates gene expression[14,19]; therefore, our data raise the possibility that FABO regulates gene expression in the rim. The apicobasal location of rim nuclei varies rapidly with the cell cycle, on the timescale of a few hours. Moreover, gene expression changes take a few hours to occur; therefore, the impact of histone acetylation on transcription will average over time. Thus, histone acetylation resulting from FABO will affect transcription in a broader domain of cells in the rim region. Selecting H3K18ac as an illustrative histone acetylation mark, we asked where FABO-dependent H3K18ac is located in the genome. We performed Cut&Run[20] in wing discs treated with or without etomoxir. Principal component analysis showed clustering of the two sample groups (Extended Data Fig. 8j). We identified 9,621 H3K18ac peaks, which were located as expected mostly in promoter

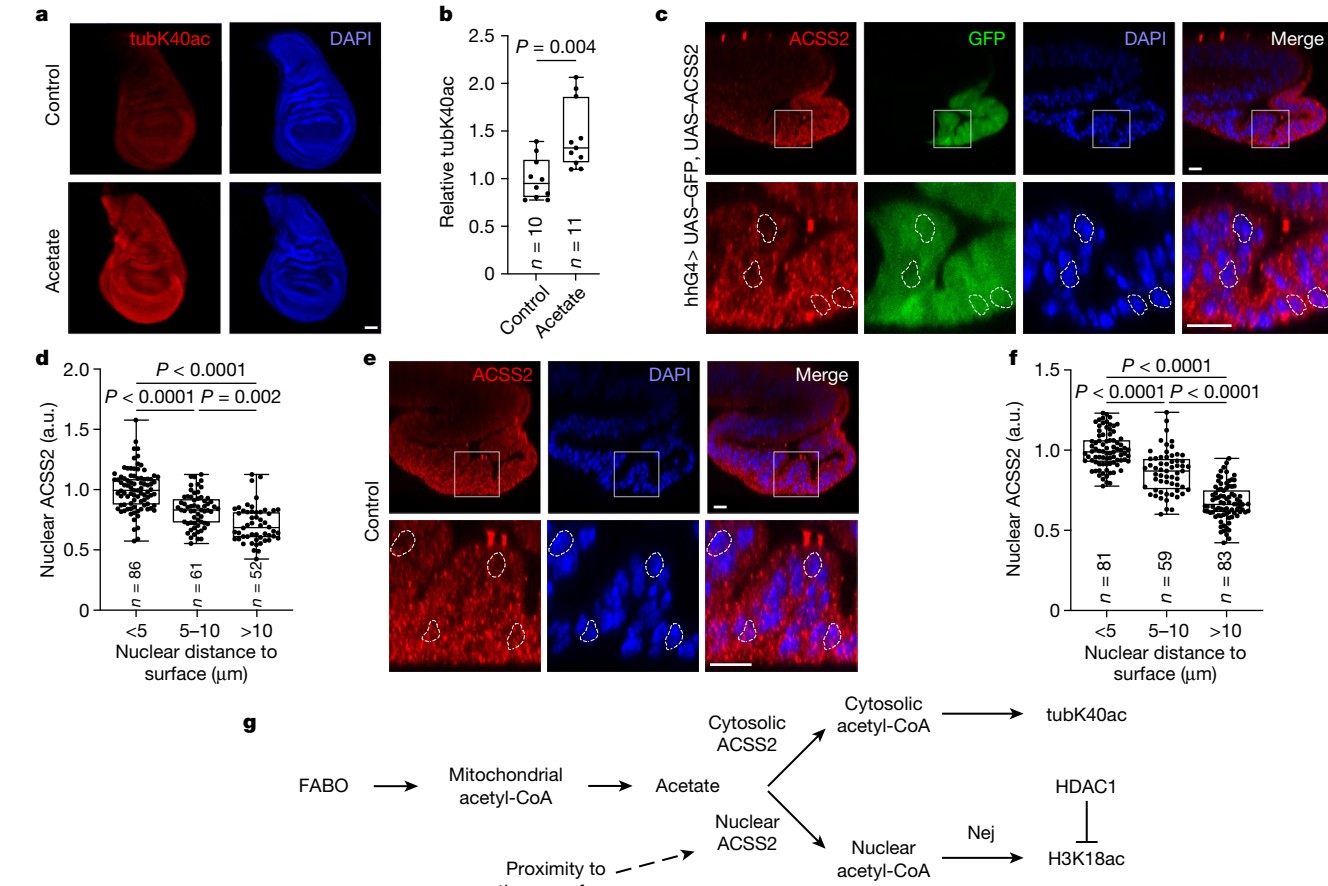

**Fig. 6 | Nuclear acetyl-CoA is produced mainly in outward-facing nuclei.**
**a**,**b**, Exogenous acetate penetrates the entire wing disc, as shown by increased uniform acetylation of tubK40ac after acetate treatment (10 mM) for 2 h in explant cultures. Representative images of sum projections in **a** and quantified in **b**. Significance by Mann–Whitney test (two-sided). **c**,**d**, Nuclear levels of overexpressed ACSS2 are higher in nuclei facing the outside of the disc tissue and lower in nuclei facing the tissue interior. The white boxes in the top panels indicate the zoomed-in areas in the bottom panels. Representative images in **c** (*n* = 14 discs) and quantified in **d**. Significance by Kruskal–Wallis test with Dunn's multiple comparisons test. Dotted shapes show the location of nuclei selected for comparison across panels. **e**,**f**, Nuclear levels of endogenous ACSS2 are higher in nuclei facing the outside of the disc tissue and lower in nuclei facing the tissue interior. The white boxes in the top panels indicate the zoomed-in areas in the bottom panels. Representative images in **e** (*n* = 14 discs) and quantified in **f**. Significance by Kruskal–Wallis test with Dunn's multiple comparisons test. **g**, Schematic diagram showing the regulation of H3K18ac in the wing disc. Box plots show the centre line (median), box limits (first and third quartiles) and whiskers (outer data points). Scale bars, 50 μm (**a**) or 10 μm (**c**,**e**).

regions[14,21] (Extended Data Fig. 8k). A total of 2,194 peaks decreased following FABO inhibition (Extended Data Fig. 8l). Gene ontology analysis of the corresponding 1,469 genes revealed enrichment for genes involved in development and developmental signalling (Extended Data Fig. 9a,b). Genes with nearby H3K18ac peaks that decreased after etomoxir treatment included proximally expressed genes, such as *mirror* (*Mirr*), *Sox box protein 15* (*Sox15*) and *homothorax* (*Hth*), among others (Extended Data Fig. 9c–j). We asked whether proper expression of these genes depends on rim histone acetylation. Efficient inhibition of FABO by etomoxir is technically challenging in larvae, and etomoxir treatment and *Nej* knockdown have similar effects, both reducing total bulk lysine acetylation (Fig. 2f and Extended Data Fig. 5c–c',f). Therefore, we knocked down *Nej* in the posterior compartment and assessed transcript levels by fluorescence in situ hybridization (FISH). Whereas expression of the housekeeping gene *glyceraldehyde 3 phosphate dehydrogenase 2* (*Gapdh2*) was not affected (Extended Data Fig. 9k–k'), expression of genes with etomoxir-dependent H3K18ac peaks was strongly downregulated in the knockdown compartment (IFP⁺; Extended Data Fig. 9c'–j', arrowheads). Hence, histone acetylation in the proximal region is required for proper gene expression.

We asked whether altered histone acetylation affects disc development. Strong knockdown of *Nej* or *Hdac1* with hhG4 is lethal to pupae. We titrated down *Nej* knockdown strength by using the weaker

engrailed-GAL4 driver at 18 °C, which produced a reduction in H3K18ac similar or weaker to that caused by loss of FABO (compare Extended Data Fig. 10a,b with Fig. 4e,f). This method generated adults with small, abnormal wings (Extended Data Fig. 10g–i'). Likewise, mild knockdown of *Hdac1* (Extended Data Fig. 10c,d) produced increased H3K18ac in the rim and caused either pupal lethality or notched wings (Extended Data Fig. 10j–j'), a result consistent with known effects on Notch signalling[22]. We also used a previously described method[23] to generated *Drosophila* in which H3K18 and H4K8 were simultaneously mutated to either non-acetylatable arginine residues or to glutamine, thereby mimicking the loss of positive charge caused by acetylation. Both mutations caused lethality (0 viable homozygotes for >1,000 animals analysed), which indicated that the regulation of acetylation on these sites is developmentally important.

Finally, we asked whether histone acetylation in the rim is specific for wing discs or a general feature of imaginal discs. Indeed, we saw high H3K18ac on the rim of all imaginal discs we inspected (Extended Data Fig. 10e,f).

## Discussion

In this study, we showed that metabolism is not uniform in the *Drosophila* wing imaginal disc, with high FABO in the rim. The acetyl-CoA

derived from FABO is used in rim nuclei to acetylate histones, which leads to increased H3K18ac, H4K8ac and total lysine acetylation, thereby affecting gene expression. Nuclei with high histone acetylation are close to the wing disc surface. This pattern did not correlate with a gene expression region or a cell cycle phase, but rather with proximity of the nucleus to the haemolymph. Nuclei cycle their positions rapidly, within 12-h cell cycles, which implies that they acetylate their histones when they approach the tissue surface and then deacetylate them when moving back deeper into the tissue. Although FABO is increased in the rim region, this is not sufficient to restrict high histone acetylation to this region because exogenously supplied acetate also specifically increased histone acetylation in the rim. Instead, nuclei close to the tissue surface have more nuclear ACSS2, which converts acetate into acetyl-CoA. We currently do not know why surface nuclei have more nuclear ACSS2, and further studies will be required to elucidate this mechanism. It will also be interesting to study whether similar phenomena occur in other organisms or in tumours, which have irregular tissue architecture and aberrant histone acetylation.

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

# Methods

## *Drosophila* stocks and clone induction in the wing disc

Fly stocks were maintained at 25 °C with a 12 h light–dark cycle. A list of fly stocks is provided in Supplementary Table 2. Detailed genotypes of all animals for all figure panels are given in Supplementary Table 3. Crosses for temperature-sensitive knockdown experiments using UAS/GAL4/tubulinG80[ts] were maintained at 18 °C and shifted to 29 °C to induce transgene expression. Crosses used for clone-induction experiments were kept at 18 °C before heat shock for 15-45 min at 35 °C (for Rok and Koi clones) or for 15–25 min at 33 °C (all other clones) during early to mid-third instar larval stages. After heat shock, crosses were kept at 25 °C to facilitate efficient gene expression. Crosses to determine the effect of mild *Nej* or *Hdac1* knockdown on disc and wing growth using enG4 were kept at 18 °C to reduce RNAi efficiency and prevent lethality.

## Immunofluorescence staining of *Drosophila* wing discs

Unless indicated otherwise, immunostainings were performed on wing discs from third instar larvae. Larvae were either dissected in PBS on ice, or incubated in explant cultures (see below)[24], and thereafter fixed for 20 min in 4% formaldehyde (FA) and PBS. All subsequent washing and incubation steps were performed with gentle rocking of the carcasses on a rocking platform. Samples were rinsed twice with PBS containing 0.3% Triton X-100 (PBX) followed by 2 washes for 10 min in PBX. Larvae were blocked for 45 min in PBX containing 5% BSA (BBX) before incubation with primary antibodies and Alexa Fluor 488 Phalloidin (Invitrogen, A12379) or Alexa Fluor 633 Phalloidin (Invitrogen, A22284) in BBX overnight at 4 °C. A list of primary and secondary antibodies used in this study is provided in Supplementary Table 4. To remove the primary antibodies, samples were rinsed twice with BBX, followed by 2 washing steps for 60 and 30 min in BBX. The samples were then incubated in BBX containing DAPI (2.5 µg ml$^{-1}$; AppliChem, A1001) and secondary antibodies (1:250) at room temperature in the dark. Before mounting the wing discs in mounting medium (80% glycerol, 1× PBS and 20 mM *n*-propyl gallate (Sigma-Aldrich, P3130)), the samples were rinsed twice and then washed 2 times for 30 min each with BBT. Immunofluorescence signals in the tissue were detected by confocal microscopy (Leica SP8 confocal microscope). Sum intensity projections were generated using Fiji.

To align wing disc orientations in the figure panels, some disc images were rotated. The resulting empty corners of the squares were underlaid with solid black boxes for a more consistent panel layout.

## Image quantification, data analysis and software

All quantifications were performed using Fiji. For quantification of histone acetylation and total ac-K immunosignals, integrated intensities were normalized to the corresponding DAPI signal and displayed as the ratio of signal of the rim versus the centre of the wing disc. The rim was defined as a region of 10 pixels in thickness starting from the edge of the disc and moving inwards. For the quantification of H3K18ac in Extended Data Fig. 10a,b, a total intensity of the anterior and posterior rim was determined rather than the ratio of rim to centre signal. For quantification of the TMRM signal, integrated intensities were normalized to the corresponding mitoGFP signal and displayed as the ratio of signal of the rim versus the centre of the wing disc. The rim was defined as a region of 13 pixels in thickness starting from the edge of the disc and moving inwards. For quantification of the tubK40ac immunosignal, sum projections were generated for tubK40ac and DAPI images in Fiji and used to measure integrated intensities. Intensities of tubK40ac were normalized to the corresponding DAPI signals. For quantification of the FISH signal, Fiji was used to measure integrated intensities, which were normalized to the corresponding DAPI signals. To combine data from biological replicates, intensities of each replicate were normalized to the average control intensity. To compare intensities or intensity ratios (rim versus centre) between anterior and posterior compartments of the same wing disc, the GFP or the IFP signal was used to identify the compartment boundaries.

For quantification of nuclear ACSS2, H3K18ac and H3 immunosignals, the DAPI signal was used to outline the nucleus, and mean nuclear intensities were measured in this region with Fiji. Additionally, the distance between the most outward-facing part of each nucleus and the outward-facing tissue surface or between the nuclear surface and the surface of the closest mitochondrion were determined. For quantification of nuclear ACSS2 immunosignal in ACSS2-overexpressing discs and control discs (Fig. 6c–f), the values were additionally sorted into three bins (<5 µm, 5–10 µm and >10 µm) and normalized to the average of the <5 µm bin.

Individual data points are given and displayed as whisker plots. Significance was determined by Mann–Whitney test (two-sided), by Kruskal–Wallis test with Dunn's multiple comparisons test, by two-way ANOVA with Šídák's multiple comparisons test or Wilcoxon signed-rank test as indicated in the respective figure legends. Pearson correlation coefficient was used to determine correlation of data. Presence of outliers was determined using ROUT.

Data were analysed using Microsoft Excel (v.16 for Mac) or GraphPad Prism (v.9)

## FISH

The FISH probes were designed to encompass between 200 and 450 nucleotides of the desired gene transcripts (see Supplementary Table 5 for the sequences of primers used). Addition of a T7 promoter sequence (ccggtaatacgactcactataggg) to the 5′ end of the reverse primer allowed for transcription directly from PCR-amplified fragments using a digoxigenin RNA labeling kit (Roche, 11277073910) according to the manufacturer's protocol. RNA probes were purified (RNA Clean-up, Macherey–Nagel, 740948) and stored at −20 °C in 50% formamide until used.

For FISH, wandering third instar larvae were dissected in PBS and fixed for 30 min in 4% FA–PBS. Following a rinse in PBS containing 0.1% Tween-20 (PBT), larvae were washed twice for 10 min and 20 min in PBT. Larvae were incubated for 5 min in PBT containing 50% methanol before transferring them to methanol, in which they could be stored for several days at −20 °C. After 5 min incubation in PBT containing 50% methanol, larvae were fixed again for 20 min in 4% FA–PBT. FA was removed by three washes in PBT (each 5 min). The wash buffer was changed to hybridization solution (HS) (50% formamide, 5× SSC (0.75 M sodium chloride and 75 mM sodium citrate dehydrate), 50 µg ml$^{-1}$ heparin and 0.1% Tween-20) by serial wash steps of 5 min each in HS–PBT dilutions of 30/70, 50/50, 70/30 and 100/0 (v/v). A 10 min wash in HS was followed by 2 h of blocking at 65 °C in HS supplemented with 100 µg ml$^{-1}$ salmon sperm DNA (Invitrogen, 15632-011). In parallel, 15 ng of FISH probe was denatured in 100 µl blocking buffer for 3 min at 80 °C. After 5 min on ice, the probe was added to the samples and incubated overnight at 65 °C. To remove unbound probe, samples were rinsed once and washed twice for 5 min and 15 min in HS. The buffer was changed back to PBT through serial wash steps of 5 min each in HS–PBT dilutions of 70/30, 50/50 and 30/70 (v/v). Subsequently, larvae were rinsed and washed 3 times for 15 min each in PBT before blocking for 60 min in maleic acid buffer (1 M maleic acid, 1.5 M NaCl; pH 7.5) supplemented with 0.5% (w/v) blocking reagent for nucleic acid hybridization and detection (Roche, 11096176001). Samples were incubated overnight at 4 °C with pre-absorbed (2–3 h incubation with larvae in the absence of FISH probe) anti-digoxigenin Fab fragments conjugated to horseradish peroxidase (Roche, 11207733910). Unbound Fab fragments were removed by 3 rinses and a 10 min wash in PBT. Nuclei were counterstained for 15 min in PBT containing DAPI (2.5 µg ml$^{-1}$), followed by a wash for 10 min in PBT. To visualize localization of the FISH probes, a TSA Plus Fluorescein kit (Akoya Biosciences, NEL741001KT) was used according to the manufacturer's protocol. Last, samples were rinsed once and washed twice for 15 min each with

PBT before mounting the wing discs in mounting medium. Fluorescent signals in the tissue were detected by confocal microscopy (Leica SP8 confocal microscope). Sum intensity projections were generated using Fiji.

To align wing disc orientations in the figure panels, some disc images were rotated. The resulting empty corners of the squares were underlaid with solid black boxes for a more consistent panel layout.

## Live staining of MMP and lipid droplets in *Drosophila* wing discs

For live imaging of the MMP using TMRM, wing discs from third instar larvae ubiquitously expressing mitoGFP driven by tubulinG4 (tubG4) were dissected in explant medium (see below)[24] and transferred into an explant chamber containing the same medium supplemented with TMRM (50 nM; Biomol, ABD-22221) and the indicated inhibitors. The concentration and duration of drug treatments are indicated in the corresponding figure legends. Supplementary Table 6 lists the compounds used in this study. After incubation in the dark, wing discs were mounted on slides in explant medium and imaged using a Leica SP8 confocal microscope.

The genetically encoded mitochondrial pH Sensor Sypher3s-dmito[15] allows visualization of the mitochondrial pH and serves as an indicator of the MMP as it is established through a proton gradient. SypHer3s-dmito is based on the hydrogen peroxide sensor Hyper, which consists of a circularly permuted yellow fluorescent protein (YFP) inserted into the regulatory domain of the bacterial hydrogen peroxide sensor OxyR, which features a mutation rendering the sensor insensitive to hydrogen peroxide but retaining sensitivity towards pH[25]. The sensor has two excitation maxima, with emission around 405 nm being pH-insensitive, whereas emission around 488 nm shows increased intensity with higher pH.

To generate SypHer3s-dmito transgenic flies, the sensor-coding sequence was transferred from a mammalian expression vector (Addgene, 108119)[15] into the pUAST-attB vector[26], allowing site-directed insertion of the transgene into the fly genome and expression under control of the UAS/GAL4-system (see Supplementary Table 5 for primers used). The transgene was inserted into the VK33 docking site by phiC31-mediated recombination.

For live imaging of the MMP using SypHer3s-dmito, wing discs from third instar larvae expressing the sensor under control of tubG4 were dissected in explant medium (see below)[24] and transferred into an explant chamber containing the same medium and the indicated inhibitors. The concentration and duration of drug treatments are indicated in the corresponding figure legends, and a list of compounds is provided in Supplementary Table 6. After incubation in the dark, wing discs were mounted on slides in PBS and imaged using a Leica SP8 confocal microscope. Images are shown in ratiometric configuration as a ratio of emission at 488 nm excitation (pH-sensitive) to 405 nm excitation (pH-insensitive). Darker colours (blue-violet) indicate lower pH and lower MMP, whereas brighter colours (yellow-white) represent higher pH and higher MMP.

Lipid droplets in the wing disc were visualized using BODIPY staining. For this purpose, wing discs from third instar larvae were dissected in explant medium (see below)[24] and transferred to explant chambers containing the same medium supplemented with BODIPY (10 μM; Sigma-Aldrich, 790389) and Hoechst 33342 (10 μg ml$^{-1}$; Sigma-Aldrich, B2261). After incubating for 10 or 30 min in the dark, larvae were rinsed once with PBS before fixation for 20 min in 4% FA. Before mounting the wing discs in mounting medium (80% glycerol, 1× PBS, 20 mM *n*-propyl gallate (Sigma-Aldrich, P3130)), the larvae were rinsed twice and then washed 2 times for 10 min each with PBS. Fluorescence signals in the tissue were detected by confocal microscopy (Leica SP8 confocal microscope).

To align wing disc orientations in the figure panels, some disc images were rotated. The resulting empty corners of the squares were underlaid with solid blacks for a more consistent panel layout.

## Explant cultures

Ex vivo explant cultures were used for drug treatments as well as live stainings of wing discs. Third instar larvae were dissected in explant medium (Schneider's *Drosophila* medium (Gibco, 21720024) containing 100 U ml$^{-1}$ penicillin, 100 μg ml$^{-1}$ streptomycin (Gibco, 15140122), 1.6 nM juvenile hormone (Sigma-Aldrich, 333725), 5 nM ecdysone (Enzo Life Sciences, LKT-E0813-M010), 8.3 ng ml$^{-1}$ adenosine deaminase (Roche, 10102105001) and 10 μg ml$^{-1}$ insulin (Sigma-Aldrich I0516)), which we previously showed maintains wing discs stress-free and proliferative for up to 6 h[24]. Incubations were performed in explant chambers (mesh basket in a small glass vial) containing explant medium under constant stirring for oxygenation. The concentration and duration of drug treatments are indicated in the corresponding figure legends and Supplementary Table 6 provides a list of compounds used.

## Cut&Run

To identify FABO-dependent H3K18ac peaks in the *Drosophila* genome, Cut&Run[20] was performed on wing discs of wandering third instar larvae. Cut&Run is a method to analyse DNA–protein interactions, similar to ChIP–seq. It is based on a fusion protein consisting of protein A, able to bind the primary antibody, and a micrococcal nuclease (MNase), which cuts gDNA around the binding site. These resulting DNA fragments are then isolated and sequenced.

In this study, a previously published *Drosophila*-optimized Cut&Run protocol was applied[27] (https://protocols.io/private/D6B0AD-2DC1431A513994A2A05AC59CDA). In brief, wandering third instar larvae were either incubated for 2 h in the presence or absence of 500 μM etomoxir in explant cultures or directly dissected. Wing discs were then bound to concanavalin A-coated magnetic beads (Polysciences, 86057-3) to prevent loss of the discs during subsequent wash steps on a magnetic rack. To allow antibody binding (H3K18ac, 1:500), wing discs were permeabilized with digitonin. After overnight incubation, bound antibodies were decorated with the protein A–MNase fusion protein (Cell Signaling, 40366). DNA digestion was induced by addition of calcium to activate MNase. Samples were supplemented with yeast spike-in DNA. Released DNA fragments were purified using AmpureXP beads (Agencourt, A63880).

Libraries were prepared using a KAPA Hyper Prep kit (Roche, 7962312001) following the manufacturer's protocol. In deviation from the protocol, KAPA Pure Beads were replaced by AmpureXP beads. Owing to low sample input in Cut&Run, adapter ligation (TruSeq single indexed DNA adapter (Illumina, 20015960)) was performed for 3 h. The PCR protocol was altered according to a previously published method[28] to shorten the PCR cycles (10 s at 60 °c) favouring exponential amplification of shorter DNA fragments produced during Cut&Run. For efficient removal of the primer peak from the library, a second post-amplification clean-up was included in the protocol, as suggested by the manufacturer. Sequencing (125 nucleotide paired-end reads, HiSeq2000, Illumina) was performed by the DKFZ Genomics and Proteomics Core Facility.

The galaxy server platform[29] was used for data analysis. Adapter sequences were trimmed using Trim Galore! (v.0.6.3) (https://github.com/FelixKrueger/TrimGalore) before aligning read sequences to the *Drosophila* genome (dm6) by Bowtie2 (v.2.4.2)[30,31]. Bowtie2 settings were adjusted according to previously published method[28] as follows: --local --very-sensitive-local --no-unal --no-mixed --no-discordant --phred33 -I 10 -X 700. Next, read duplicates were removed using MarkDuplicates (v.2.18.2.2) (http://broadinstitute.github.io/picard/). Peak calling was performed using MACS2 callpeak (v.2.1.1.20160309.6)[32,33]. Finally, differential binding and principal component analysis was assessed using DiffBind (v.2.10.0)[34] by grouping the samples according to treatment. Peaks were annotated using ChIPseeker (v.1.18.0)[35] with the genome annotation file from Ensembl (dm6, genes and gene

prediction). Gene ontology enrichment analysis was performed using the online tool http://www.webgestalt.org.

## Generation of fly lines expressing mutant histones

Fly lines expressing mutant histones were generated using a previously published system[23]. An entry vector for Gateway Cloning (Thermo Fisher Scientific) provided by A. Herzig encoding a histone gene unit (His-GU) was used as a template to simultaneously mutate H3K18 and H4K8 to either H3R18 and H4R8 or to H3Q18 and H4Q8 (see Supplementary Table 5 for primers used). Cloning of the mutant His-GU into three entry vectors (using XhoI and BstBI) allowed generation of a final destination vector containing three mutated His-GUs by Gateway Cloning. This destination vector, optimized per ref. 23, contains recombination sites allowing phiC31-mediated insertion into the fly genome. The transgene was inserted separately into the VK33 and ZH86Fb docking sites. Subsequently, the two transgenic fly lines were recombined, generating a line with 6 His-GU on chromosome III and crossed into a histone null mutant background (HisC).

## Reporting summary

Further information on research design is available in the Nature Portfolio Reporting Summary linked to this article.

## Data availability

All deep-sequencing datasets are available at the NCBI Gene Expression Omnibus database (accession number GSE207486). The genome annotation we used is from Ensembl (dm6) available at http://ensembl.org. Source data are provided with this paper.

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

**Acknowledgements** We thank A. Herzig for sharing flies and plasmids for generating the histone-mutant animals; M. Mannervik for sharing the anti-Nej antibody; A. Mahmoud from S. Eaton's laboratory for sharing the anti-Lpp antibodies; F. Casares for sharing the tshG4 flies; S. Müller for embryo injections to generate transgenic animals; and staff at the DKFZ Genomics and Proteomics Core Facility (Heidelberg, Germany) for next-generation sequencing of the Cut&Run libraries.

**Author contributions** P.W. performed all experiments. P.W. and A.A.T. designed experiments, analysed data and wrote the manuscript.

**Funding** Open access funding provided by Deutsches Krebsforschungszentrum (DKFZ).

**Competing interests** The authors declare no competing interests.

**Additional information**
**Correspondence and requests for materials** should be addressed to Aurelio A. Teleman.

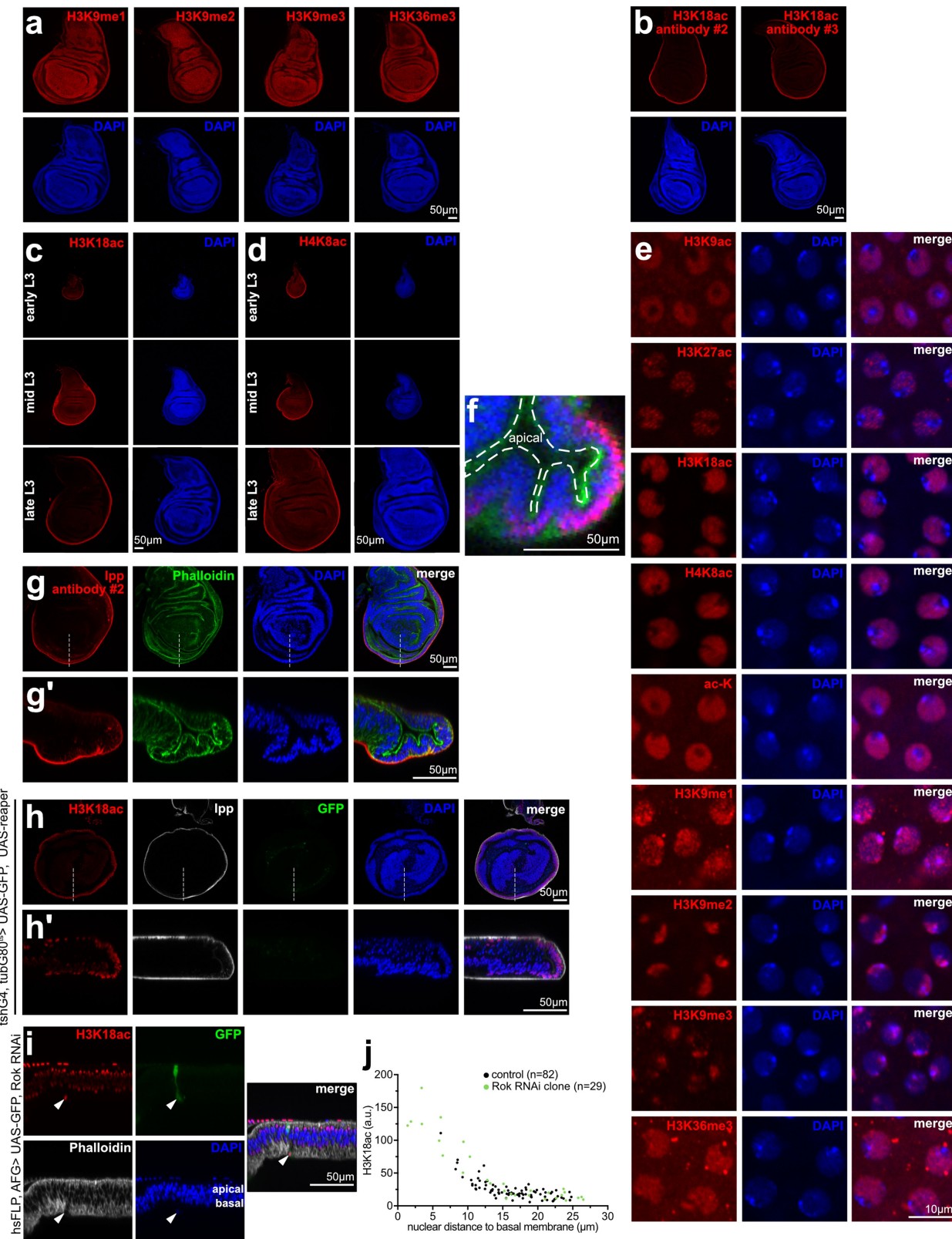

**Extended Data Fig. 1** | See next page for caption.

**Extended Data Fig. 1 | Histone methylation is uniform while histone acetylation is non-uniform in the wing disc. (a)** Histone methylation marks H3K9me1, H3K9me2, H3K9me3, and H3K36me3 show a roughly uniform distribution in the wing disc as visualized by immunostaining. Euchromatin-associated H3K36me3 is slightly enriched in the disc rim and heterochromatin-associated H3K9me2 is slightly decreased in the wing disc rim, suggesting more open chromatin in the rim. (n = 7, 14, 12, and 20 discs from left to right). **(b)** Elevated rim H3K18ac is also visible with two additional independent antibodies. (n = 19 and 20 discs from left to right). **(c-d)** H3K18ac (c) and H4K8ac (d) are enriched in the wing disc rim throughout the third-instar larval stage (early to late L3). (c, n = 14, 20, and 11 discs from left to right; d, n = 18, 17, and 11 discs from top to bottom). **(e)** Images of peripodial nuclei suggest enrichment of H3K9ac, H3K27ac, H3K18ac, H4K8ac, and H3K36me3 as well as total acetylated lysine (ac-K) in euchromatin as their immunosignals are excluded from highly compacted DNA regions indicated by high DAPI signal, whereas H3K9me2 and H3K9me3 are associated with heterochromatin, indicated by high DAPI signal. H3K9me1 is present throughout the nucleus. (n = 7, 8, 16, 12, 13, 7, 14, 12, and 20 discs from top to bottom). **(f)** Higher magnification view of main Fig. 1h where the apical surface is marked with a dotted white line. **(g-g′)** Enrichment of lipoproteins (lpp) on outward-facing membranes was validated by a second lpp antibody (#2). Dashed line in (g) indicates the position of the cross-section shown in (g′). (n = 15 discs). **(h-h′)** The formation of a new rim of H3K18ac-positive nuclei after elimination of the tsh expression domain via reaper expression correlates with high levels of lpp on outward-facing membranes. Cell death was induced by temperature-sensitive expression of reaper for 1 day at 29 °C. Dashed line in (h) indicates the position of the cross-section shown in (h′). (n = 5 discs). **(i-j)** Nuclei in Rok loss-of-function (RNAi) clones in the wing pouch move further basally than normal, where they come in proximity to the tissue surface and have high H3K18ac. Representative images in (i), quantified in (j). Nuclei from control cells and Rok loss-of-function clones are indicated in black and green, respectively. (n = 23 discs). Box plots: center line (median), box limits (1st and 3rd quartiles), whiskers (outer data points).

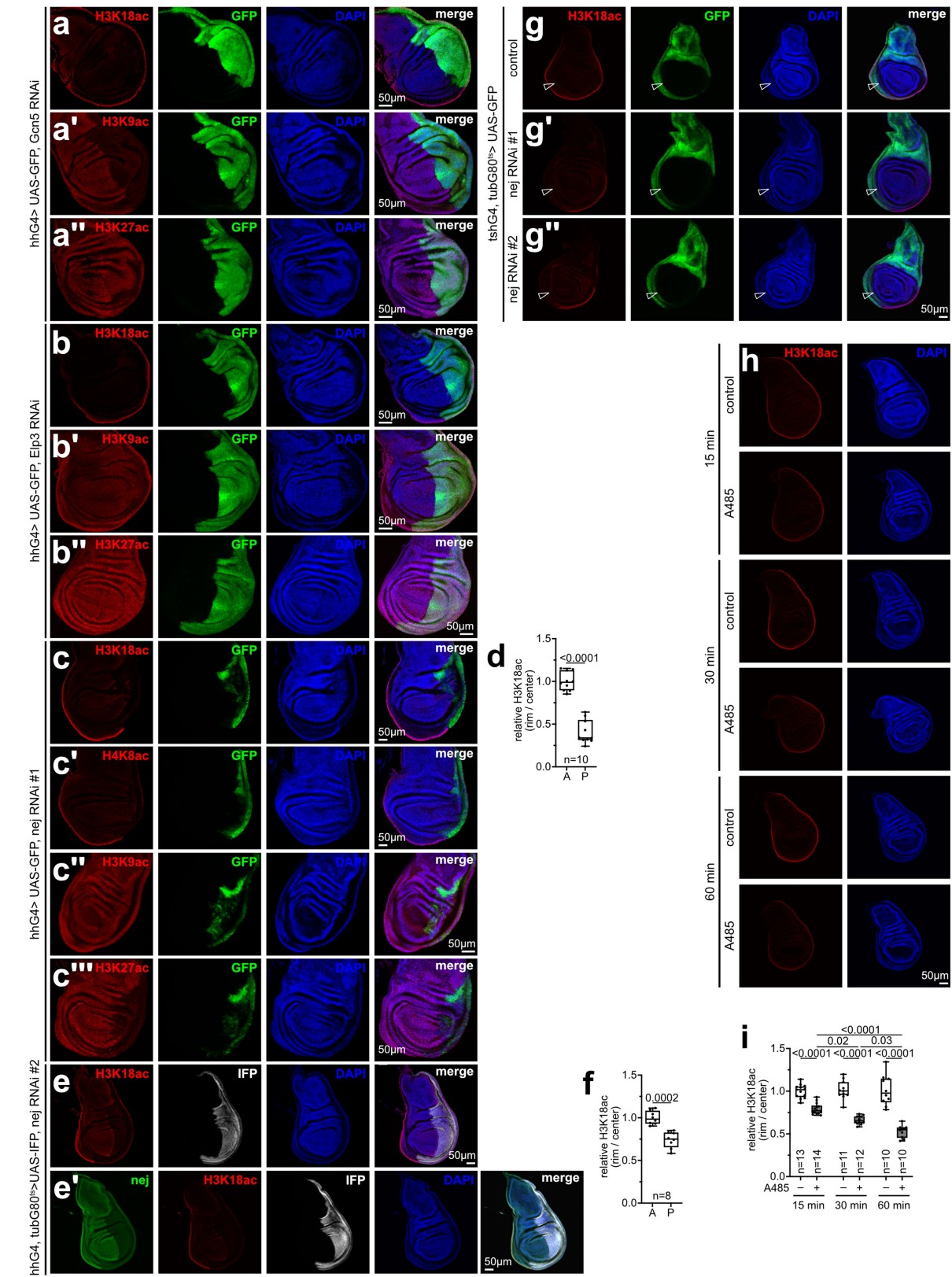

**Extended Data Fig. 2** | See next page for caption.

**Extended Data Fig. 2 | H3K18 is acetylated by nejire. (a-a")** General control non-repressed protein 5 (Gcn5) acetylates H3K9 (a') and H3K27 (a"), but not H3K18ac (a), assessed via Gcn5 knockdown in the posterior compartment (GFP+). (n = 10 discs). **(b-b")** Elongator complex protein 3 (Elp3) acetylates neither H3K18 (b), H3K9 (b') or H3K27 (b") as knockdown of Elp3 does not visibly affect levels of either histone mark in the posterior compartment (GFP+). (b, n = 9 discs; b', n = 9 discs; b", n = 16 discs). **(c-d)** nejire (nej) acetylates H3K18 (c), H4K8 (c'), and H3K27 (c"), but not H3K9ac (c"'), assessed via nej knockdown in the posterior compartment (GFP+). Representative images of H3K18ac in (c), quantified in (d). Statistical significance by Mann-Whitney test (two-sided). (c, n = 13 discs; c', n = 8 discs; c", n = 7 discs; c"', n = 10 discs). **(e-f)** Acetylation of H3K18 by nej was validated with a second independent nej RNAi line which also shows loss of H3K18ac upon knockdown in the posterior compartment (IFP+), as shown in main Fig. 2e. Temperature-sensitive knockdown of nej was induced for 1 day at 29 °C. Representative images in (e), quantified in (f). Statistical significance by Mann-Whitney test (two-sided). (e') Nej knockdown efficiency, and specificity of the nej antibody were validated using a second RNAi targeting nej. Temperature-sensitive knockdown of nej was induced for 1 day at 29 C in the posterior compartment (IFP+). (n = 10 discs). **(g-g")** Brief knockdown of nej in the proximal domain of the wing disc does not cause formation of a new interior rim. Knockdown of nej with two independent RNAi lines (g'-g") in the teashirt (tsh) domain, visualized by GFP expression, causes loss of H3K18ac, but not loss of rim tissue and no new internal H3K18ac^high rim is formed (GFP-, open arrowhead). This outcome is in stark contrast to ablation of the tsh domain with reaper (main Fig. 1f,g) which leads to loss of the tissue and formation of a new H3K18ac^high domain. Temperature-sensitive knockdown of nej was induced for only 1 day at 29 °C (n = 23 discs). **(h-i)** Pharmacological inhibition of nej with A485 (20 μM) causes a marked decrease in H3K18ac within 15 min of treatment. H3K18ac immunosignal further decreases with prolonged treatment (up to 60 min). Representative images in (h), quantified in (i). Statistical significance by two-way ANOVA with Šídák's multiple comparisons test. Box plots: center line (median), box limits (1st and 3rd quartiles), whiskers (outer data points). A, anterior; P, posterior.

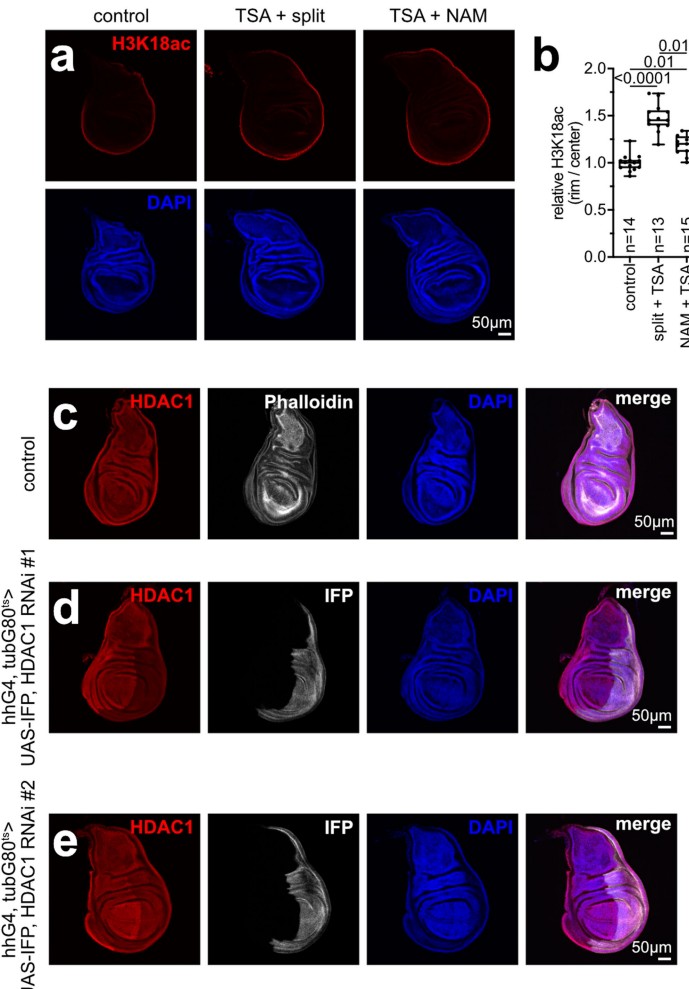

**Extended Data Fig. 3 | H3K18ac is not higher on the rim because of deacetylation in the center. (a-b)** The H3K18ac rim pattern is not due to elevated deacetylation in the interior/pouch region. Pharmacological inhibition of all histone deacetylases (sirtuins with splitomycin (split, 50 μM) or nicotinamide (NAM, 20 mM) and HDACs with trichostatin A (TSA, 500 nM)) only increases H3K18ac in the rim of the disc without changing the pattern. Discs incubated for 2 h in explant cultures with indicated inhibitors. Representative images in (a), quantified in (b). Statistical significance by Kruskal-Wallis test with Dunn's multiple comparisons test. **(c-e)** HDAC1 protein is uniformly expressed in the wing disc (c). Specificity of the antibody was validated by HDAC1 knockdown using two independent RNAi lines (d-e) causing a decrease in immunosignal. Temperature-sensitive knockdown of HDAC1 was induced for only 1 day at 29 °C in the posterior compartment (IFP+) because longer knockdown causes tissue loss. (c, n = 25 discs; d, n = 11 discs; e, n = 11 discs). Box plots: center line (median), box limits (1st and 3rd quartiles), whiskers (outer data points). A, anterior; P, posterior.

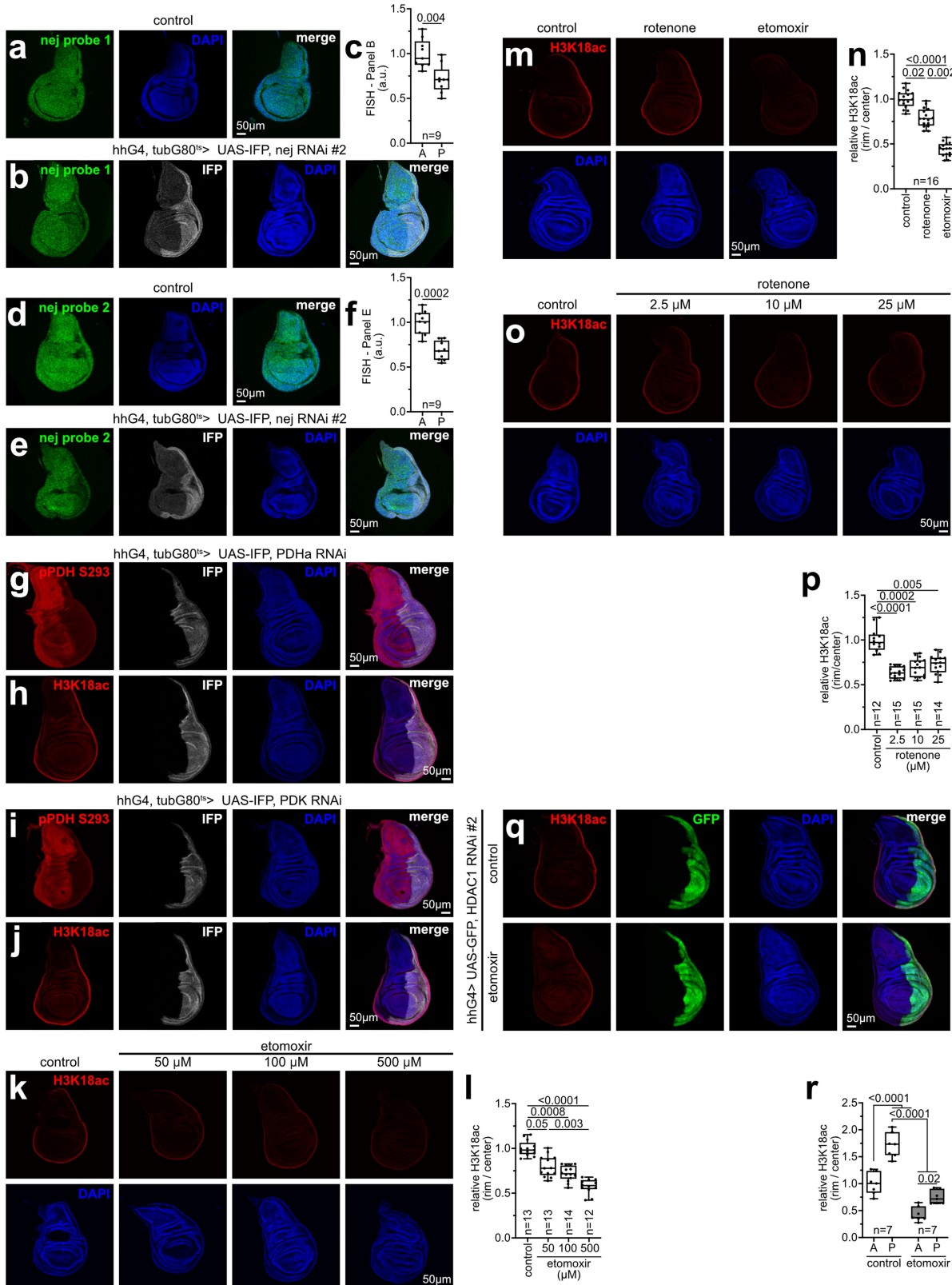

**Extended Data Fig. 4** | See next page for caption.

**Extended Data Fig. 4 | FABO is required for acetylation in the rim of the wing disc. (a-c)** Nejire (nej) transcript is uniform in the wing disc, detected by fluorescence in situ hybridization (FISH) in wildtype discs (a), or in nej knockdown discs (b) to validate specificity of the nej FISH probe. Temperature-sensitive knockdown of nej was induced for only 1 day at 29 °C because longer knockdown causes tissue loss. Representative images in (b), quantified in (c). Statistical significance by Mann-Whitney test (two-sided). **(d-f)** Uniform distribution of nejire (nej) transcript in the wing disc was validated with a second FISH probe (d). Specificity of the second nej FISH probe was validated by nej knockdown in the posterior compartment (IFP+), decreasing the nej FISH signal. Temperature-sensitive knockdown of nej was induced for 1 day at 29 °C. Representative images in (e), quantified in (f). Statistical significance by Mann-Whitney test (two-sided). **(g-j)** Glucose-derived acetyl-CoA is not required for acetylation of H3K18. Knockdown of pyruvate dehydrogenase α (PDHa, which should reduce glycolytic acetyl-CoA synthesis) or of pyruvate dehydrogenase kinase (PDK, which should increase glycolytic acetyl-CoA synthesis), each validated by pPDH S293 immunostaining (g and i, respectively), does not impact levels of H3K18ac (h and j, respectively). Temperature-sensitive knockdown of PDHa and of PDK was induced for 1 day at 29 °C in the posterior compartment (IFP+). (g, n = 9 discs; h, n = 10 discs; i, n = 9 discs; j, n = 8 discs). **(k-l)** Etomoxir-induced decrease in H3K18ac is concentration dependent. Etomoxir treatment shows already at low concentrations (50 μM) a decrease in H3K18ac levels. H3K18ac immunosignal further decreases with increasing levels of etomoxir (100 μM, 500 μM). Discs incubated with the indicated concentrations for 2 h in explant cultures. Representative images in (k), quantified in (l). Statistical significance by Kruskal-Wallis test with Dunn's multiple comparisons test. **(m-n)** Decrease in H3K18ac upon etomoxir treatment is not due to inhibition of respiratory chain complex I. Inhibition of complex I with rotenone (10 μM) only mild decreases H3K18ac whereas etomoxir treatment (500 μM) causes strong loss of the acetylation mark. Discs incubated with the indicated compounds for 2 h in explant cultures. Representative images in (m), quantified in (n). Statistical significance by Kruskal-Wallis test with Dunn's multiple comparisons test. **(o-p)** Rotenone activity is already maximal at 2.5 μM. Discs incubated for 2 h with various concentrations of rotenone (2.5-25 μM) and immunostained for H3K18ac. Representative images in (o), quantified in (p). Statistical significance by Kruskal-Wallis test with Dunn's multiple comparisons test. **(q-r)** The decrease in H3K18ac caused by inhibition of FABO (500 μM etomoxir) is blunted by HDAC1 knockdown in the posterior compartment (GFP+). Discs incubated in the presence or absence of etomoxir for 1 h in explant cultures. Representative images in (q), quantified in (r). Statistical significance by two-way ANOVA with Šídák's multiple comparisons test. Box plots: center line (median), box limits (1st and 3rd quartiles), whiskers (outer data points). A, anterior; P, posterior.

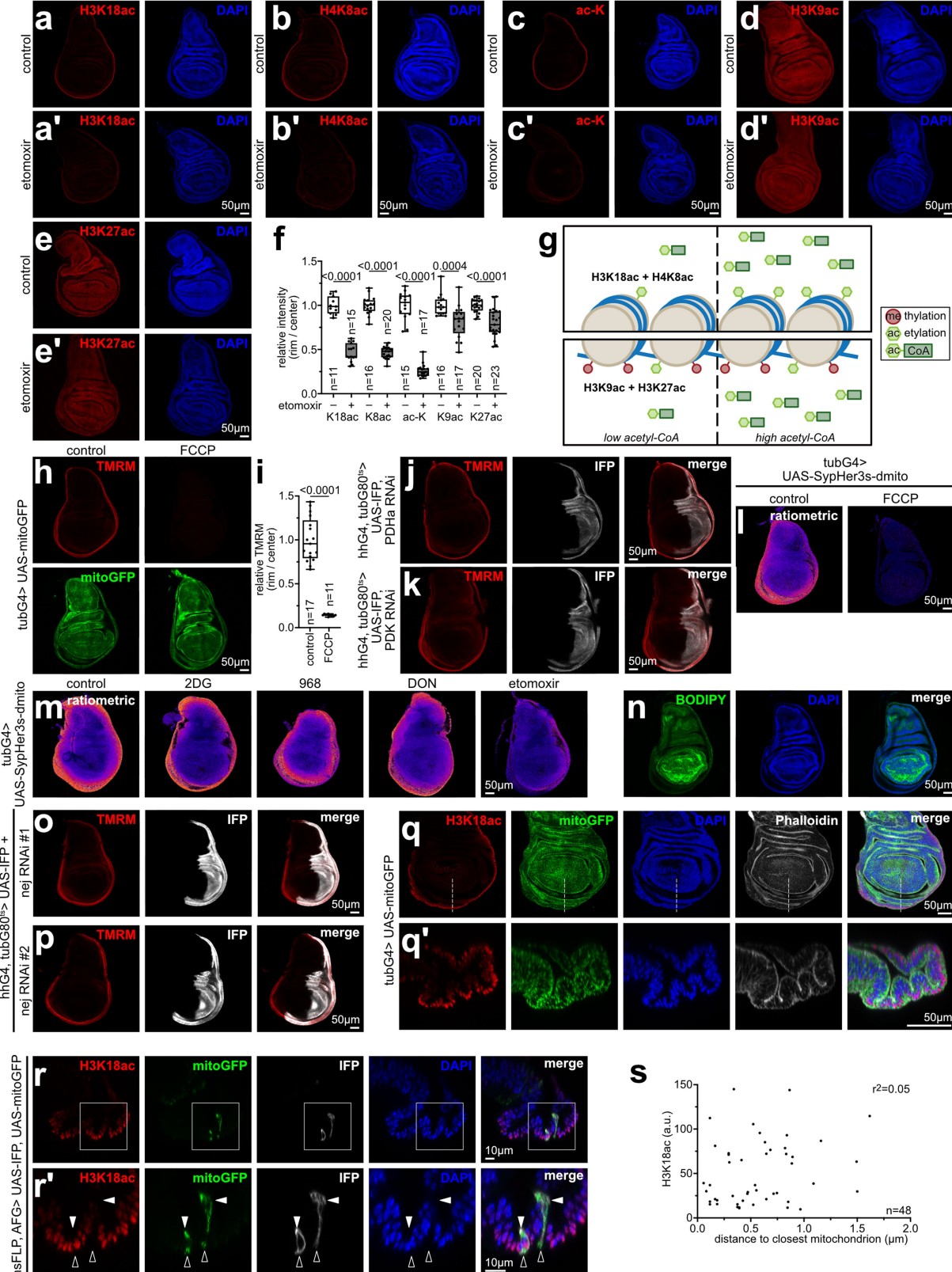

**Extended Data Fig. 5 |** See next page for caption.

**Extended Data Fig. 5 | Fatty acid β-oxidation provides acetate for histone acetylation in the rim. (a-f)** Inhibition of FABO by etomoxir (500 μM) decreases levels of acetylation in the rim of the disc as shown by immunostaining for H3K18ac (a-a'), H4K8ac (b-b'), total acetylated lysine (ac-K; c-c'), H3K9ac (d-d'), and H3K27ac (e-e'). Discs incubated in the presence or absence of etomoxir for 2 h in explant cultures. Representative images in (a-e'), quantified in (f). Statistical significance by Mann-Whitney test (two-sided). **(g)** Schematic diagram summarizing the effect of altered acetyl-CoA levels on different histone acetylation marks. For H3K9 and H3K27 which are also methylated, changes in acetyl-coA levels have little effect on acylation stoichiometry. Instead, for H3K18 and H4K8, which are not methylated, altered levels of acetyl-CoA lead to changes in acetylation levels. **(h-i)** Treatment of discs with the mobile ion carrier carbonyl cyanide-p-trifluoromethoxyphenylhydrazone (FCCP, 10 μM) causes loss of tetramethylrhodamine-methylester (TMRM) staining, validating the specificity of TMRM in visualizing the mitochondrial membrane potential. Mitochondria visualized by tubG4-driven mitoGFP expression. Discs incubated with FCCP for 30 min in explant cultures with TMRM. Representative images in (h), quantified in (i). Statistical significance by Mann-Whitney test (two-sided). **(j-k)** Glycolysis is not responsible for high mitochondrial membrane potential in the rim because knockdown of pyruvate dehydrogenase α (PDHa, j) or of pyruvate dehydrogenase kinase (PDK, k) does not impact TMRM staining. Temperature-sensitive knockdown of PDHa and of PDK was induced for 1 day at 29 °C in the posterior compartment (IFP+). Discs stained for 30 min in explant cultures with TMRM. (j, n = 11 discs; k, n = 12 discs). **(l)** Ubiquitous tubG4-driven expression of a genetically-encoded, mitochondrially-localized pH-sensor (SypHer3s-dmito) shows elevated mitochondrial membrane potential in the rim of wing discs, as also shown using TMRM in main Fig. 5a. See Methods for details on the ratiometric imaging. Specificity of the sensor was validated by dissipating the membrane potential using 10 μM FCCP. Discs incubated with FCCP for 1 h in explant cultures. (n = 4 discs from one independent experiment). **(m)** The importance of FABO in maintaining a high mitochondrial membrane potential in the rim was validated with SypHer3s-dmito. See Methods for details on the ratiometric imaging. Inhibition of FABO (500 μM etomoxir) but not of glycolysis (20 mM 2-deoxy-glucose (2DG)) or glutaminolysis (50 μM glutaminase inhibitor 968 (968) or 500 μM 6-diazo-5-oxo-L-norleucin (DON)) caused a drop in high mitochondrial membrane potential in the rim of the disc. Discs incubated with the indicated inhibitors for 1 h in explant cultures. (n = 4 discs from one independent experiment). **(n)** BODIPY staining shows enrichment of lipid droplets in the pouch, anti-correlating with high FABO in the rim of the disc. (n = 13 discs). **(o-p)** Acetylation of H3K18 is not upstream of FABO because nej knockdown using two independent RNAi lines does not impact mitochondrial membrane potential in the posterior compartment (IFP+). Temperature-sensitive knockdown of nej was induced for 1 day at 29 °C. Discs incubated for 30 min in explant cultures with TMRM. (o) n = 8 discs; (p), n = 15 discs. **(q-q')** High H3K18ac in the rim of the wing disc is not caused by a higher concentration of mitochondria in this region, since mitochondrial distribution is rather uniform throughout the wing disc. Mitochondria visualized with tubulinG4 (tubG4)-driven mitoGFP. Dashed line in (q) indicates position of the cross-section shown in (q'). (n = 13 discs). **(r-r')** Cells have apical (solid arrowheads) and basal mitochondria (open arrowheads), visualized by mitoGFP expression, independent of the nuclear position and H3K18ac levels, arguing against a regulatory role for mitochondrial position in determining the non-uniform H3K18ac pattern. The white box in (r) indicates the zoomed-in area in (r'). (n = 15 discs). **(s)** H3K18ac does not correlate with nuclear-mitochondrial distance. Using mitoGFP expressing clones, as shown in (r), nuclear distance to the closest mitochondrion of the same cell was determined and plotted against H3K18ac immunosignal. The coefficient of determination $r^2$ was used to determine what fraction of the variability of H3K18ac is explained by nuclear-mitochondrial distance. Box plots: center line (median), box limits (1st and 3rd quartiles), whiskers (outer data points).

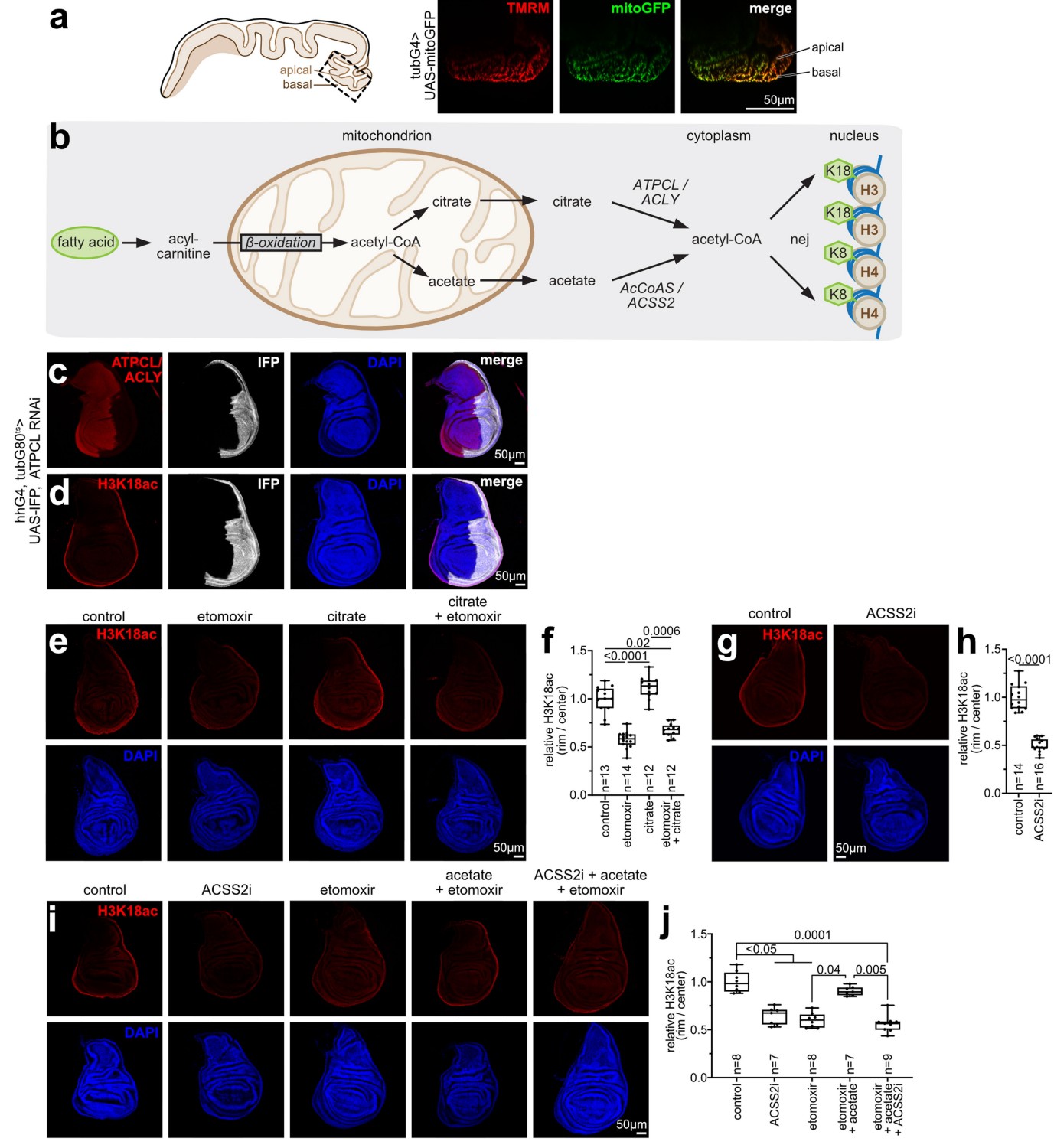

**Extended Data Fig. 6 |** See next page for caption.

**Extended Data Fig. 6 | Acetyl-CoA Synthase (ACSS2) generates acetyl-CoA for H3K18ac. (a)** Cross-sections of wing discs stained with TMRM show high mitochondrial membrane potential in the proximal folds, indicating a broader pattern than high H3K18ac. Discs incubated for 30 min in explant cultures with TMRM. (n = 13 discs). **(b)** FABO produces acetyl-CoA inside mitochondria which is converted to acetate or citrate for export from this organelle. Acetate and citrate, in turn, can be used to generate nuclear acetyl-CoA by acetyl-CoA synthase (AcCoAS/ACSS2) or ATP citrate lyase (ATPCL/ACLY). Nuclear acetyl-CoA is used by acetyltransferases such as nejire (nej) to acetylate histones. **(c-d)** ACLY is not required for H3K18ac. ACLY knockdown, validated by ACLY immunostaining (c), does not impact H3K18ac levels (d). Temperature-sensitive knockdown of ACLY was induced for 2 days at 29 °C in the posterior compartment (IFP+). (c, n = 9 discs; d, n = 11 discs). **(e-f)** Acetyl-CoA required for acetylation of H3K18 is not derived from citrate, as citrate treatment (10 mM) does not rescue loss of H3K18ac upon inhibition of FABO inhibition (500 µM etomoxir). Discs incubated with the indicated compounds for 2 h in explant cultures. Representative images in (e), quantified in (f). Statistical significance by Kruskal-Wallis test with Dunn's multiple comparisons test. **(g-h)** ACSS2 provides acetyl-CoA for H3K18ac. Inhibition of ACSS2 (20 µM ACSS2 inhibitor, ACSS2i) for 2 h in explant cultures causes loss of H3K18ac. Representative images in (g), quantified in (h). Statistical significance by Mann-Whitney test (two-sided). **(i-j)** ACSS2 is required for acetate (10 mM) to rescue the etomoxir-induced (500 µM) loss of H3K18ac. Discs incubated for 2 h in explant cultures with indicated compounds ± ACSS2i (20 µM). Representative images in (i), quantified in (j). Statistical significance by Kruskal-Wallis test with Dunn's multiple comparisons test. Box plots: center line (median), box limits (1st and 3rd quartiles), whiskers (outer data points).

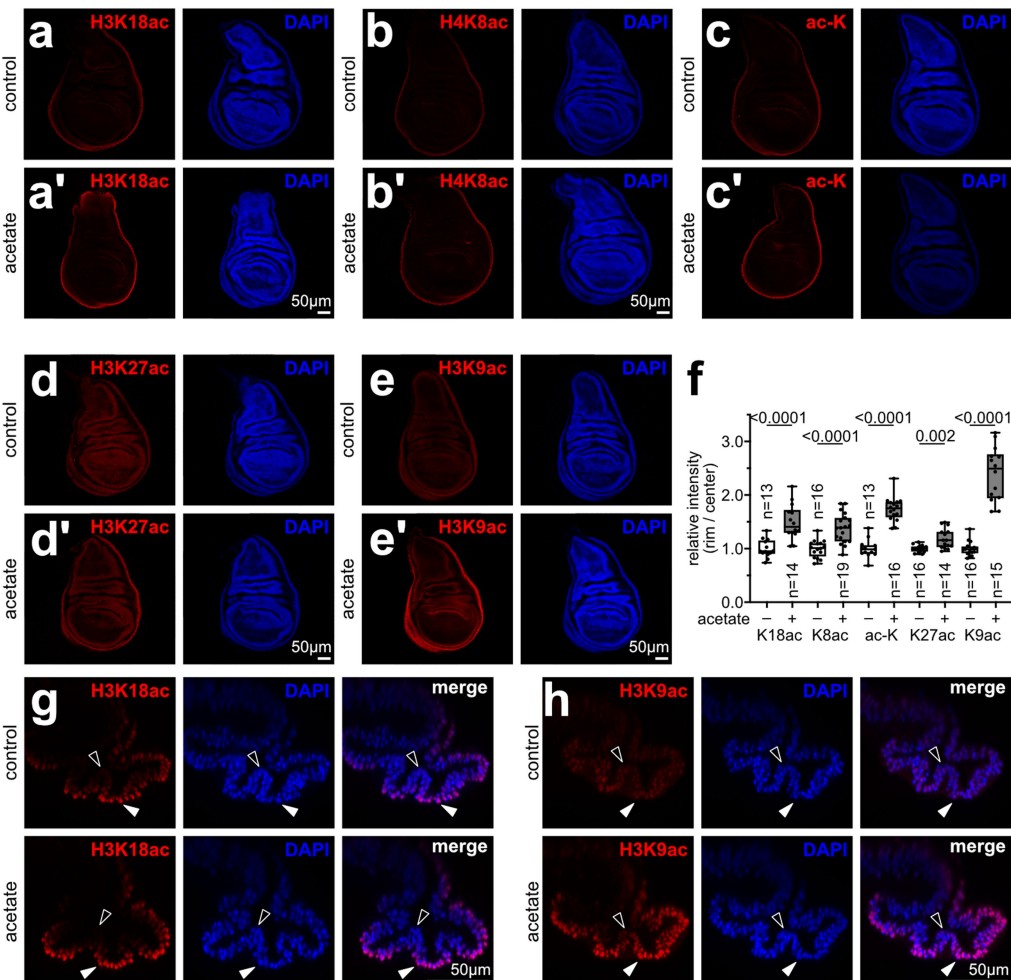

**Extended Data Fig. 7 | Incubation with acetate leads to an increase in histone acetylation in the rim of the wing disc. (a-f)** Incubation with acetate (10 mM, 2 h) causes a general increase in histone acetylation in the rim of the wing disc but not in the pouch of the wing disc. Shown are H3K18ac (a-a'), H4K8ac (b-b'), total acetylated lysine (ac-K; c-c'), H3K27ac (d-d'), and H3K9ac (e-e'). Representative images in (a-e), quantified in (f). Statistical significance by Mann-Whitney test (two-sided). **(g-h)** Acetate treatment only increases acetylation of histones in outward-facing nuclei. Cross-sections of acetate-treated (10 mM, 2 h) discs immunostained for H3K18ac (g) or H3K9ac (h) show increased histone acetylation in nuclei facing the outside of the tissue (solid arrowheads) but not the tissue interior (open arrowheads). (g, n = 12 and 10 discs from top to bottom; h, n = 12 and 14 discs from top to bottom). Box plots: center line (median), box limits (1st and 3rd quartiles), whiskers (outer data points).

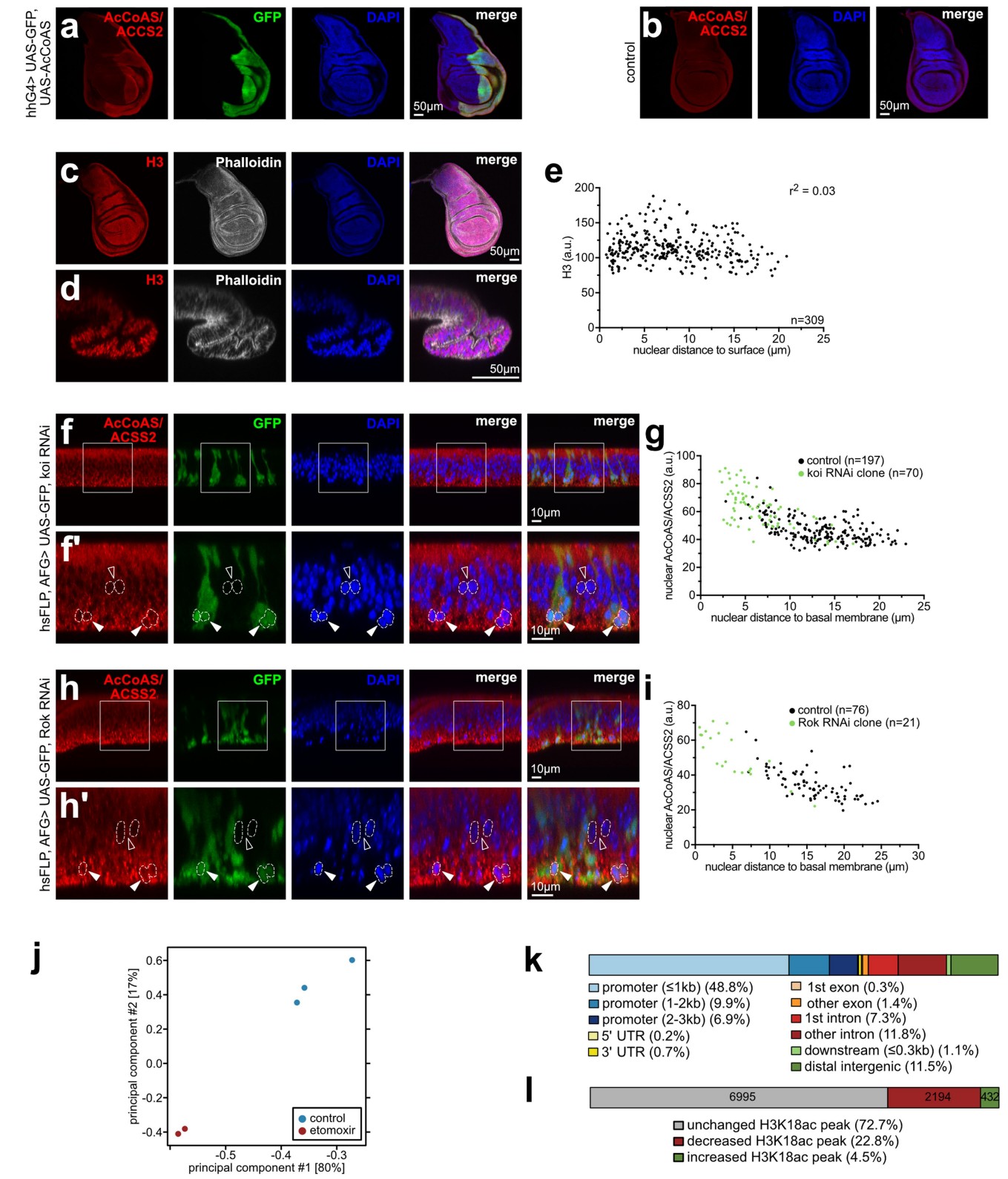

**Extended Data Fig. 8** | See next page for caption.

**Extended Data Fig. 8 | Nuclear position regulates levels of nuclear ACSS2.**
**(a)** Specificity of the acetyl-CoA synthase (ACSS2) antibody was validated by overexpressing ACSS2 in the posterior compartment (GFP+). (n = 14 discs). **(b)** Uniform distribution of endogenous ACSS2 protein in the wing disc detected by immunostaining. (n = 14 discs). **(c-e)** As a negative control for main Fig. 6c–f, signal for immunostaining of another nuclear protein, histone 3 (H3), do not depend on distance of the nucleus to the tissue surface. X-Y section shown in (c) and cross-section in (d). (e) Quantification of H3 immunosignal versus nuclear distance to the surface. The coefficient of determination $r^2$ was used to determine what fraction of the variability of H3 staining is explained by distance of the nucleus to the tissue surface. (c-d, n = 17 discs). **(f-g)** Nuclei of koi mutant clones in the wing pouch which move more basally than usual, and hence in proximity to the tissue surface, have elevated levels of nuclear ACSS2. Representative clones are shown in f-f′. The white box in (f) indicates the zoomed-in area in (f′). (g) Quantification of nuclear ACSS2 levels versus distance of each nucleus from the basal surface of the tissue. Nuclei from koi RNAi clones are shown in green, control nuclei in black. (f-f′, n = 15 discs). **(h-i)** Nuclei of Rok mutant clones in the wing pouch which move more basally than usual, and hence in proximity to the tissue surface, have elevated levels of nuclear ACSS2. Representative clones are shown in h-h′. The white box in (h) indicates the zoomed-in area in (h′). (i) Quantification of nuclear ACSS2 levels versus distance of each nucleus from the basal surface of the tissue. Nuclei from Rok RNAi clones are shown in green, control nuclei in black. (h-h′, n = 14 discs). **(j)** Principal component analysis of Cut&Run samples reveals clear clustering of control (blue) and etomoxir-treated samples (red). **(k)** H3K18ac peaks identified by Cut&Run are mainly located in promoter regions (66%). **(l)** Inhibition of FABO by etomoxir (500 μM, 2 h) decreases H3K18ac on a specific set of peaks (red).

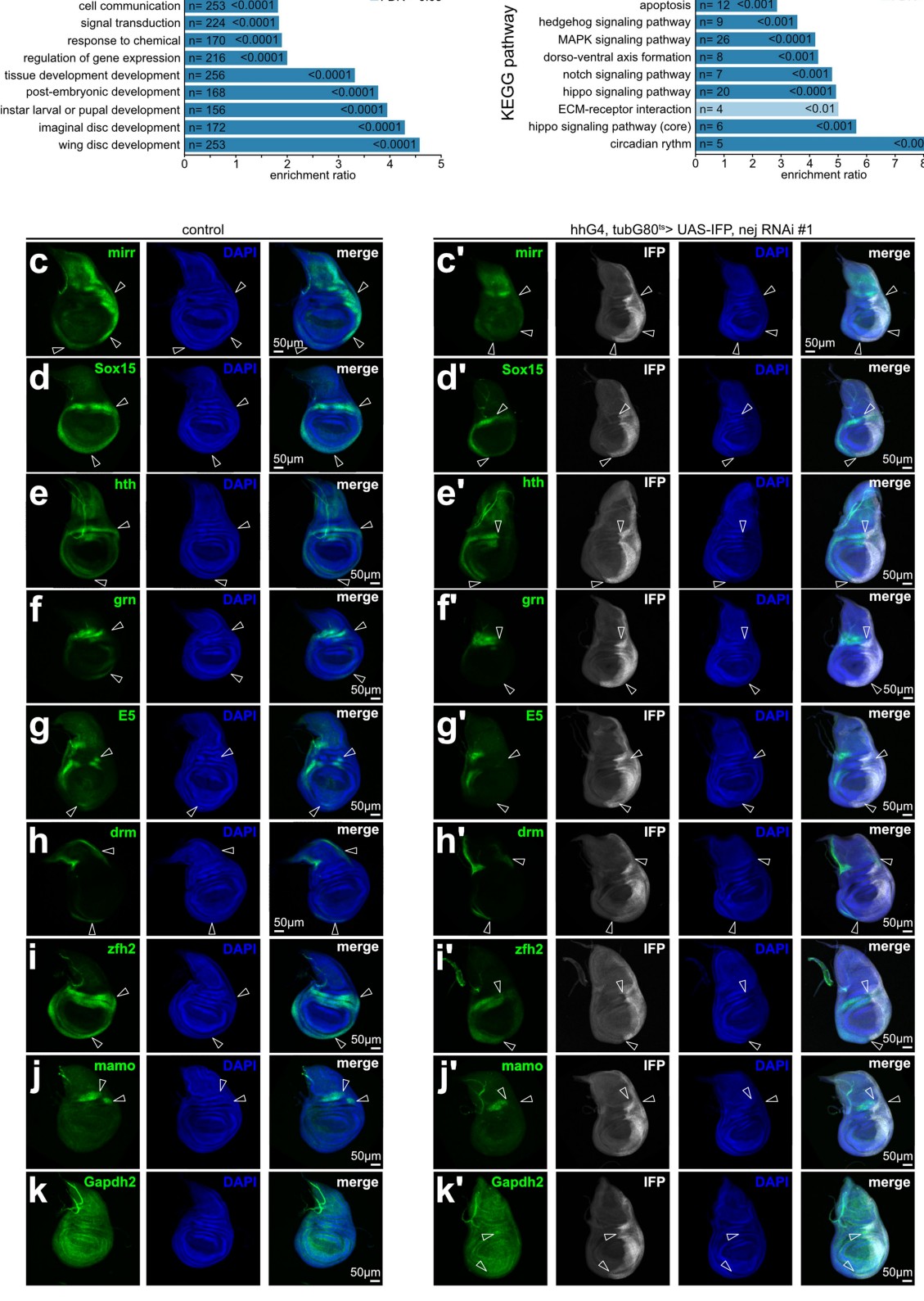

**Extended Data Fig. 9 | Rim histone acetylation regulates gene expression in the wing disc. (a-b)** GO enrichment on the set of genes with reduced H3K18ac upon FABO inhibition from Extended Data Fig. 8l. FDR, false discovery rate. **(c-k')** Some genes with H3K18ac peaks dependent on FABO (as identified in Extended Data Fig. 8l) have a proximal expression pattern that requires nejire (nej)-dependent histone acetylation, as exemplified by fluorescence in situ hybridization (FISH) for mirror (mirr; c-c'), Sox box protein 15 (Sox15; d-d'), and homothorax (hth; e-e'), and other (f-j'), whereas house-keeping genes such as glyceraldehyde-3-phosphate dehydrogenase 2 (Gapdh2; k-k') do not. Temperature-sensitive knockdown of nej was induced for 1.5 days at 29 °C in the posterior compartment (IFP+). Images shown as sum projections. (c-k, n = 9, 8, 8, 8, 9, 8, 7, 8, and 9 discs, respectively; c'-j', n = 11, 9, 10, 10, 10, 11, 10, and 10 discs, respectively; k', n = 4 discs from one independent experiment).

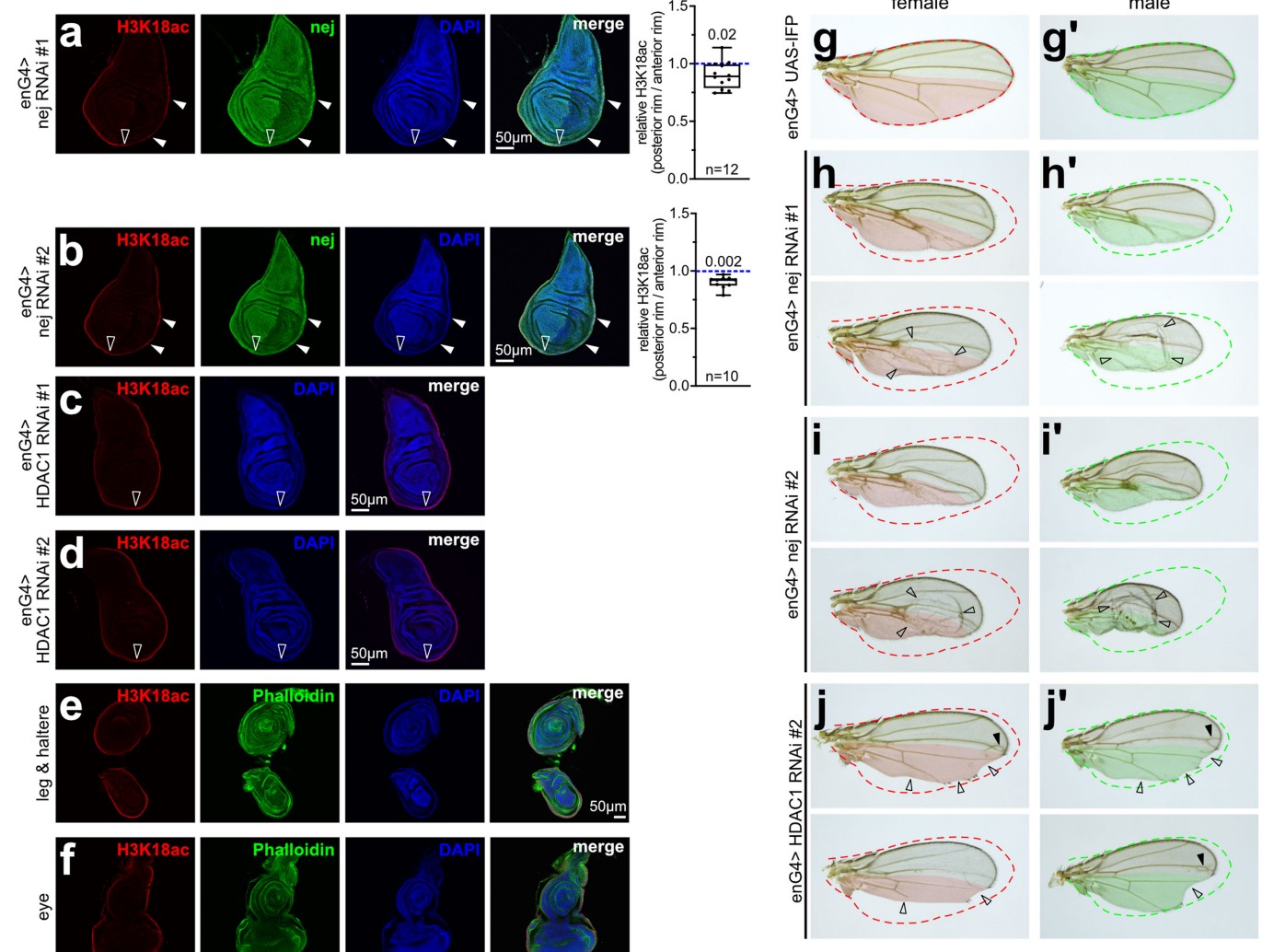

**Extended Data Fig. 10 | Correct regulation of rim histone acetylation is required for proper wing development. (a-b')** Mild knockdown of nej in the posterior compartment using two independent RNAi lines, causes a decrease in nej and H3K18ac immunosignals. Patches of remaining H3K18ac in the posterior rim co-localize with higher levels of residual nej protein (solid arrowhead). The anterior-posterior border is indicated by an open arrowhead. Flies were kept at 18 °C to reduce RNAi efficiency in order to prevent pupal lethality. Representative images on the left, quantified on the right. Statistical significance by Wilcoxon signed-rank test (two-sided). (b') 1 outlier was removed as determined by ROUT. **(c-d)** Mild knockdown of HDAC1 in the posterior compartment using two independent RNAi lines, causes a mild increase in H3K18ac immunosignal. The anterior-posterior border is indicated by an open arrowhead. Flies were kept at 18 °C to reduce RNAi efficiency in order to prevent pupal lethality. (c, n = 12 discs; d, n = 18 discs). **(e-f)** Other imaginal discs such as the leg and haltere imaginal discs (e) or the eye imaginal disc (f) also have high H3K18ac on their outer rim, similar to wing discs. (e, n = 15 discs; f, n = 14 discs). **(g-j')** Knockdown of nej and HDAC1 causes growth defects of the wing. Mild knockdown of nej in the posterior compartment using two independent RNAi lines (h-i') was achieved with enG4 at 18 C, leading to decreased wing size compared to wings of control flies (g-g'). Some wings also display blisters (open arrowhead). Mild knockdown of HDAC1 in the posterior compartment causes wing notches (j-j'; open arrowheads) as well as defects in vein patterning (solid arrowhead). Flies were kept at 18 °C to reduce RNAi efficiency in order to prevent pupal lethality. Wings of female and male flies are shown in (g-j) and (g'-j'), respectively. Outline of control wings is indicated by dotted lines and the posterior compartment is colored. (g, n = 59 wings; g', n = 60 wings; h, n = 70 wings; h', n = 58 wings; I, n = 74 wings; i', n = 58 wings; j-j', n = 60 wings).

# Reporting Summary

## Statistics

For all statistical analyses, confirm that the following items are present in the figure legend, table legend, main text, or Methods section.

| n/a | Confirmed | |
|---|---|---|
| ☐ | ☒ | The exact sample size (*n*) for each experimental group/condition, given as a discrete number and unit of measurement |
| ☐ | ☒ | A statement on whether measurements were taken from distinct samples or whether the same sample was measured repeatedly |
| ☐ | ☒ | The statistical test(s) used AND whether they are one- or two-sided *Only common tests should be described solely by name; describe more complex techniques in the Methods section.* |
| ☒ | ☐ | A description of all covariates tested |
| ☐ | ☒ | A description of any assumptions or corrections, such as tests of normality and adjustment for multiple comparisons |
| ☐ | ☒ | A full description of the statistical parameters including central tendency (e.g. means) or other basic estimates (e.g. regression coefficient) AND variation (e.g. standard deviation) or associated estimates of uncertainty (e.g. confidence intervals) |
| ☐ | ☒ | For null hypothesis testing, the test statistic (e.g. *F*, *t*, *r*) with confidence intervals, effect sizes, degrees of freedom and *P* value noted *Give P values as exact values whenever suitable.* |
| ☒ | ☐ | For Bayesian analysis, information on the choice of priors and Markov chain Monte Carlo settings |
| ☒ | ☐ | For hierarchical and complex designs, identification of the appropriate level for tests and full reporting of outcomes |
| ☐ | ☒ | Estimates of effect sizes (e.g. Cohen's *d*, Pearson's *r*), indicating how they were calculated |

*Our web collection on statistics for biologists contains articles on many of the points above.*

## Software and code

Policy information about availability of computer code

| Data collection | The software running the Leica confocal microscope was Leica Application Suite X (LAS X) version 3.5.2.18963, which is commercially available. |
|---|---|
| Data analysis | All software used in this study is publicly or commercially available:<br>Images were analyzed with FIJI.<br>Cut&Run sequencing data sets were analyzed using the galaxy server platform (https://usegalaxy.org)<br>- Adapter sequences were trimmed using Trim Galore! (version 0.6.3) (https://github.com/FelixKrueger/TrimGalore)<br>- Reads were aligned using Bowtie2 (version 2.4.2).<br>- Read duplicates were removed using MarkDuplicates (version 2.18.2.2) (http://broadinstitute.github.io/picard/).<br>- Peak calling was performed by MACS2 callpeak (version 2.1.1.20160309.6)<br>- Differential binding was evaluated using DiffBind (version 2.10.0)<br>- Peaks were annotated by ChIPseeker (version 1.18.0) using the genome annotation file from Ensembl (dm6, genes and gene prediction).<br>GO enrichment analysis was performed with the online tool http://www.webgestalt.org.<br>Figure preparation: Affinity Photo and Affinity Designer.<br>Data analysis: Microsoft Excel or Graphpad Prism. |

For manuscripts utilizing custom algorithms or software that are central to the research but not yet described in published literature, software must be made available to editors and reviewers. We strongly encourage code deposition in a community repository (e.g. GitHub). See the Nature Portfolio guidelines for submitting code & software for further information.

## Data

All manuscripts must include a data availability statement. This statement should provide the following information, where applicable:
- Accession codes, unique identifiers, or web links for publicly available datasets
- A description of any restrictions on data availability
- For clinical datasets or third party data, please ensure that the statement adheres to our policy

> All deep sequencing datasets are available at NCBI Geo (accession GSE207486). Reviewer token: irklausybhoztqx

## Human research participants

Policy information about studies involving human research participants and Sex and Gender in Research.

| | |
|---|---|
| Reporting on sex and gender | Not applicable. |
| Population characteristics | Not applicable. |
| Recruitment | Not applicable. |
| Ethics oversight | Not applicable. |

Note that full information on the approval of the study protocol must also be provided in the manuscript.

# Field-specific reporting

Please select the one below that is the best fit for your research. If you are not sure, read the appropriate sections before making your selection.

☒ Life sciences      ☐ Behavioural & social sciences      ☐ Ecological, evolutionary & environmental sciences

For a reference copy of the document with all sections, see nature.com/documents/nr-reporting-summary-flat.pdf

# Life sciences study design

All studies must disclose on these points even when the disclosure is negative.

| | |
|---|---|
| Sample size | No sample size calculations were performed. Sample sizes were determined based on practical considerations (e.g. number of animals available, processing time required) and expected variations between animals for a given sample. Based on these considerations, sample sizes were set at >6 animals per genotype/condition, which allowed modest standard deviations and significant p values to be obtained. |
| Data exclusions | One outlier (as determined by ROUT) was removed from ED Fig. 10b'. Otherwise, no data were excluded. |
| Replication | Except where noted in the figure legends, experiments were replicated at least 2 times. Only data that reproduced are included in this manuscript. |
| Randomization | All animals were randomly alloted. |
| Blinding | The experiments were not blinded because they each contained an internal control and because blinding is not usual in this field. |

# Reporting for specific materials, systems and methods

We require information from authors about some types of materials, experimental systems and methods used in many studies. Here, indicate whether each material, system or method listed is relevant to your study. If you are not sure if a list item applies to your research, read the appropriate section before selecting a response.

## Materials & experimental systems

| n/a | Involved in the study |
|---|---|
| ☐ | ☒ Antibodies |
| ☒ | ☐ Eukaryotic cell lines |
| ☒ | ☐ Palaeontology and archaeology |
| ☐ | ☒ Animals and other organisms |
| ☒ | ☐ Clinical data |
| ☒ | ☐ Dual use research of concern |

## Methods

| n/a | Involved in the study |
|---|---|
| ☐ | ☒ ChIP-seq |
| ☒ | ☐ Flow cytometry |
| ☒ | ☐ MRI-based neuroimaging |

# Antibodies

**Antibodies used**

antibody source species source  product number  LOT
AcCoAS/ACSS2 rabbit Abcam Ab264391 GR3350114-1
ac-K rabbit Cell Signaling 9814S 5
ATPCL/ACLY rabbit Novus Biologicals NBP1-90266 R09359
H3 rabbit Cell Signaling 2650S 3, 4
H3K9ac rabbit Active Motif 39137 9811002
H3K9me1 rabbit Abcam Ab9045 GR323589-1
H3K9me2 mouse Abcam Ab1220 GR183500-3
H3K9me3 rabbit Abcam Ab8898 GR3176468-1
H3K18ac rabbit Abcam Ab1191 GR300534-1, GR3287957-1
H3K18ac #2 rabbit Cell Signaling 9675 2
H3K18ac #3 rabbit Active Motif ACM-39756 14722003
H3K18crot rabbit Cusabio CSB-PA010 418OA18crHU G0822A
H3K27ac rabbit Abcam Ab4729 GR323154
H3K36me3 rabbit Abcam Ab9050 GR166781-1
H4K8ac rabbit Abcam Ab15823 GR3209076-1, GR3294416-1
HDAC1 rabbit ProteinTech 10197-1-AP 106660
lpp  guinea pig Eaton Lab [Eugster, 2007, https://doi.org/10.1016/j.devcel.2007.04.019]
lpp #2 rabbit Eaton Lab [Eugster, 2007, https://doi.org/10.1016/j.devcel.2007.04.019]
nejire guinea pig Mannervik lab [Holmqvist, 2012, https://doi.org/10.1371/journal.pgen.1002769]
pH3 S10 mouse  Cell Signaling 9706S 17
pPDH S293 rabbit Abcam Ab92696 GR319281-1
tubK40ac rabbit Cell Signaling 5335 6
anti-rabbit TRITC donkey Jackson ImmunoResearch 711-025-152
anti-mouse TRITC donkey Jackson ImmunoResearch 715-025-150
anti-mouse FITC donkey Jackson ImmunoResearch 715-095-151
anti-guinea pig FITC goat Jackson ImmunoResearch 106-095-003
anti-guinea pig Cy5 donkey Jackson ImmunoResearch 706-175-148

**Validation**

The lpp antibodies from the laboratory of Suzanne Eaton were validated in the following publication:
PMID: 17609110 and the nejire antibody from the laboratory of Mattias Mannervik was validated in the following publication: PMID: 22737084.

The following antibodies were  validated in this study by gene knockdown or over-expression: AcCoAs/ACSS2, ATPCL/ACLY, HDAC1, nejire, pPDH S293.

AcCoAS/ACSS2, Abcam, Ab264391: validated https://www.abcam.com/products/primary-antibodies/acss2-antibody-ab264391.html
ac-K, Cell Signaling, 9814S: validated https://www.cellsignal.com/products/primary-antibodies/acetylated-lysine-ac-k-2-100-multimab-rabbit-mab-mix/9814
ATPCL/ACLY, Novus Biologicals, NBP1-90266 : validated https://www.novusbio.com/products/atp-citrate-lyase-antibody_nbp1-90266
H3, Cell Signaling, 2650S: validated https://www.cellsignal.com/products/primary-antibodies/histone-h3-antibody-chip-formulated/2650
H3K9ac, Active Motif ,39137: validated https://www.activemotif.com/catalog/details/39137/histone-h3-acetyl-lys9-antibody-pab
H3K9me1, Abcam, Ab9045: validated https://www.abcam.com/products/primary-antibodies/histone-h3-mono-methyl-k9-antibody-chip-grade-ab9045.html
H3K9me2, Abcam, Ab1220: validated https://www.abcam.com/products/primary-antibodies/histone-h3-di-methyl-k9-antibody-mabcam-1220-chip-grade-ab1220.html
H3K9me3, Abcam, Ab8898: validated https://www.abcam.com/products/primary-antibodies/histone-h3-tri-methyl-k9-antibody-chip-grade-ab8898.html
H3K18ac, Abcam, Ab1191: validated https://www.abcam.com/products/primary-antibodies/histone-h3-acetyl-k18-antibody-chip-grade-ab1191.html
H3K18ac #2, Cell Signaling, 9675: validated https://www.cellsignal.com/products/primary-antibodies/acetyl-histone-h3-lys18-antibody/9675
H3K18ac #3, Active Motif, ACM-39756: validated https://www.activemotif.com/catalog/details/39755/histone-h3-acetyl-lys18-antibody-pab-3
H3K18crot, Cusabio, CSB-PA010 418OA18crHU: validated https://www.cusabio.com/Polyclonal-Antibody/Crotonyl-HIST1H3A--K18--Antibody-12784039.html
H3K27ac, Abcam, Ab4729: validated https://www.abcam.com/products/primary-antibodies/histone-h3-acetyl-k27-antibody-chip-

grade-ab4729.html
H3K36me3, Abcam, Ab9050: validated https://www.abcam.com/products/primary-antibodies/histone-h3-tri-methyl-k36-antibody-chip-grade-ab9050.html
H4K8ac, Abcam, Ab15823: validated https://www.abcam.com/products/primary-antibodies/histone-h4-acetyl-k8-antibody-chip-grade-ab15823.html
HDAC1, ProteinTech, 10197-1-AP: validated https://www.ptglab.com/products/HDAC1-Antibody-10197-1-AP.htm
pH3 S10, Cell Signaling, 9706S: validated https://www.cellsignal.com/products/primary-antibodies/phospho-histone-h3-ser10-6g3-mouse-mab/9706
pPDH S293, Abcam, Ab92696: validated https://www.abcam.com/products/primary-antibodies/pdha1-phospho-s293-antibody-ab92696.html
tubK40ac, Cell Signaling, 5335: validated https://www.cellsignal.com/products/primary-antibodies/acetyl-a-tubulin-lys40-d20g3-xp-174-rabbit-mab/5335

# Animals and other research organisms

Policy information about studies involving animals; ARRIVE guidelines recommended for reporting animal research, and Sex and Gender in Research

| Laboratory animals | Species: Drosophila melanogaster<br>Strains: Full genotype for each figure panel is provided in Suppl. Table 3.<br>Age: wL3 |
|---|---|
| Wild animals | This study did not involve wild animals. |
| Reporting on sex | Sex was generally not considered in the study design - both males and females were analyzed together. Only for the analysis of adult wings in ED Figure 10g-j' are wings from males and females shown separately. |
| Field-collected samples | This study did not involve samples collected from the field. |
| Ethics oversight | This study does not require an ethical approval. |

Note that full information on the approval of the study protocol must also be provided in the manuscript.

# ChIP-seq

## Data deposition

☒ Confirm that both raw and final processed data have been deposited in a public database such as GEO.

☒ Confirm that you have deposited or provided access to graph files (e.g. BED files) for the called peaks.

| Data access links<br>*May remain private before publication.* | https://www.ncbi.nlm.nih.gov/geo/query/acc.cgi?acc=GSE207486 |
|---|---|
| Files in database submission | GSM6290457 (wing disc control)<br>GSM6290458 (wing  disc control  2h BR1)<br>GSM6290459 (wing disc control 2h BR2)<br>GSM6290460 (wing disc + etomoxir BR1)<br>GSM6290461 (wing disc + etomoxir BR2)<br>GSE207486_Galaxy24_ChIPseeker...  (peak calling) |
| Genome browser session<br>(e.g. UCSC) | http://genome.ucsc.edu/s/willnow/CUT%26RUN_submission<br>note: control 2h BR2 was split into 2 tracks due to size. |

## Methodology

| Replicates | control: 3 biological replicates<br>treatment: 2 biological replicates |
|---|---|
| Sequencing depth | control: total reads = 61 Mio, unique = 5.2 Mio<br>control 2h BR1: total reads = 45 Mio, unique = 4.3 Mio<br>control 2h BR2: total reads = 67 Mio, unique = 47.6 Mio<br>etomoxir BR1: total reads = 57 Mio, unique = 4.7 Mio<br>etomoxir BR2: total reads = 72 Mio, unique = 4.3 Mio<br><br>reads: paired-end<br>sequence length: 101 bp |
| Antibodies | H3K18ac rabbit Abcam Ab1191 GR300534-1, GR3287957-1 |

| Peak calling parameters | Peak calling was performed using the galaxy server plattform using MACS2 callpeak (version 2.1.1.20160309.6, standard setting). |
|---|---|
| Data quality | Read duplicates were removed using the galaxy server plattform using MarkDuplicates (version 2.18.2.2) . Peak calling used a FDR cut off of 5%. |
| Software | The galaxy server platform 25 was used for data analysis. Adapter sequences were trimmed using Trim Galore! (version 0.6.3) (https://github.com/FelixKrueger/TrimGalore) before aligning read sequences to the Drosophila genome (dm6) by Bowtie2 (version 2.4.2). Bowtie2 settings were adjusted according to Skene et al. as follows: --local --very-sensitive-local --no-unal --no-mixed --no-discordant --phred33 -I 10 -X 700. Next, read duplicates were removed using MarkDuplicates (version 2.18.2.2) (http://broadinstitute.github.io/picard/). Peak calling was performed by MACS2 callpeak (version 2.1.1.20160309.6). Finally, differential binding was evaluated using DiffBind (version 2.10.0) by grouping the samples according to treatment. Peaks were annotated by ChIPseeker (version 1.18.0) using the genome annotation file from Ensembl (dm6, genes and gene prediction). |

