## [Peer Review File · Nature]

Manuscript Title: Nuclear position and local acetyl-CoA production regulate chromatin state

Reviewer Comments & Author Rebuttals

Reviewer Reports on the Initial Version:

Referees' comments:

Referee #1 (Remarks to the Author):

This paper shows that specific histone acetylation marks (principally H3K9ac and H4K16ac) are selectively present in nuclei of cells comprising the rim of the *Drosophila* wing disc, a developmental model of a pseudo-stratified epithelium. Using a combination of genetic and pharmacological manipulations along with localization of specific histone acetylation marks, the authors build a case that selective acetylation of histones results from the specific access of nuclei in the rim to substrate Ac-CoA derived from mitochondrial produced metabolites. By perturbing proteins implicated in nuclear positioning, the LINC complex protein klaroid (*koi*) and the general myosin II contractility regulator Rho kinase, they argue that the selective labeling of histone in nuclei at the periphery depends on the position of the nucleus and not other factors.

Overall, the paper contains novel and interesting results on the selective marking of histones in nuclei of a pseudo-stratified epithelia, and intricate sleuthing to identify the proximal metabolic substrate that provides the source of Ac-CoA for the specific histone marks. Together these results point to the novel possibility that nuclear positioning may contribute to alterations in the epigenome. Yet, there are significant issues that need to be resolved before the paper is advanced to publication in a general interest journal such as *Nature*.

I have three major concerns with the overall conclusion of the paper that the physical position of the nucleus determines its repertoire of histone marks and hence chromatin state:

1. First, the perturbations used to alter nuclear position (lof for *koi* and Rho kinase) both also generally affect mechanotransduction to the nucleus. Thus, as far as the authors have taken it, it is not possible to distinguish whether these lof perturbations are solely affecting nuclear position rather than other nuclear mechanical properties such as stiffness, chromatin compaction, or even nuclear/cytoplasmic

transport, each of which may separately affect histone acetylation. Perhaps the authors will argue that probing such nuclear mechanical properties will entail studying mechanisms that they have not yet delved into, but without discerning amongst them, there is really no way to support the conclusion reflected in the title of the paper.

2. Second, I am not convinced by the data presented that proximity of the nucleus to mitochondria does not contribute to the selective histone marks. The observation that mitochondria are present throughout the epithelia does not rule out the possibility that mitochondria are in much closer proximity to nuclei in the periphery. High magnification images and measurements of actual nuclear-mitochondrial distances are necessary.

3. Three, roughly half of the paper (Figures 3-5; ED Figures 6-8) comprises results identifying the Ac-CoA carbon source and enzymes. Yet, this aspect of the paper is not reflected in the title of the paper. This seems to be an oversight to me. If it turns out nuclear positioning is involved, isn't an equally important part of the story that there is localized generation of the substrate for the histone acetylases? To me the current title is not fully reflective of the results.

Other comments:

4. Key to the authors' conclusion about nuclear positioning is their interpretation that in *koi* (or *Rjo* kinase) *lof* cells with nuclei displaced toward the periphery, the nuclei are positive for H3K18ac. They assume that the cells were negative for H3K18ac before *koi/Rho* kinase *lof*, yet it was unclear to me how they know this, particularly as there are many adjacent nuclei in non-*koi* expressing cells that are H3K18ac positive. Additionally, quantification is needed as only single examples are shown in Fig. 1k and ED Fig 1i.

5. In general, higher magnification images are needed to make the following results more convincing: a) the position of nuclei relative to the apical-basal aspect of the epithelium (Fig. 1g'; here it would also be useful to have apical and basal markers); b) the proximity of nuclei to mitochondria (ED Fig. 7h-h'), and c) whether the ACSS2 puncta are actually nuclear (Fig 5e-f, g-h).

6. Some of the conclusions about loss/gain of signal are anecdotal and would benefit from more quantification (e.g., ED Fig 4a-b'''; Fig 3b, ED Fig 6b,g)

7. A caveat to the conclusion that high deacetylation does not occur in the interior is that there may be more than one deacetylase(s) (in addition to HDAC1) contributing to the signal in the interior.

8. The rescue by octanoate in Fig. 3g,h is partial and this should be indicated in the text. Similarly with HDAC1 kd in ED Fig. 6g.

Referee #2 (Remarks to the Author):

This study by Willnow and Teleman shows a new link between fatty acid oxidation (FAO) and acetyl-CoA for histone acetylation in a spatially restricted manner. While the direct link between FAO and histone acetylation is known, this study shows that histone acetylation is required for proper gene expression and development in the outer rim disc, but not other parts of the *Drosophila* wing disc. Generally, this study is well-controlled and provides a logical flow between hypothesis and experiment. I can only offer one major critique:

MAJOR:

The study's biggest limitation is that the authors don't know how proximity to the tissue surface influences nuclear ACSS2 localization and generation of substrate for NEJ. They convincingly show that fatty acid β -oxidation is high in the rim of the disc, and that this region also has more (generally?) nuclear ACSS2 localization. Both of these events are required for histone acetylation and gene expression, but the authors don't know *how* this occurs. The key idea and novelty of the manuscript are in how the physical position of a nucleus within a tissue affects its chromatin state. But as presented, the authors don't know the mechanism(s) explaining how positioning influences metabolism, chromatin, and gene expression.

MINOR:

Beyond inhibiting CPT1, Etomoxir at high concentrations consumes free CoA and disrupts many metabolic pathways. Genetic ablation of CPT1 would further strengthen the evidence for FAO.

Referee #3 (Remarks to the Author):

In the manuscript, Willnow et al. report higher levels of total lysine acetylation, H3K18ac and H4K8ac marks in the outer rim of the *Drosophila* wing discs. H3K9ac, H3K27ac and some methylation marks (H3K9me1/2/3, H3K27me3) studied do not show the same enrichment. Nejire is identified as the main acetyltransferase responsible for the deposition of the H3K18ac mark. However, nej mRNA levels are uniform in the wing disc. Unlike inhibition of glycolysis and glutaminolysis, inhibition of fatty acid β -oxidation via etomoxir treatment decreases the H3K18ac levels in the rim. This suggests that β -oxidation is the main source of acetyl-CoA in the rim. The authors claim there is higher β -oxidation in the rim based on the higher mitochondrial membrane potential observed, which is sensitive to etomoxir treatment. In addition, higher nuclear levels of the enzyme ACSS2 are reported in the wing disc rim. The authors suggest that the conversion of acetate to acetyl-CoA by the nuclear ACSS2 population is the rate limiting step in moderating H3K18ac levels in the wing disc rim. Finally with a H3K18ac Cut & Run experiment with and without the treatment of the wing discs with etomoxir, the authors identify the H3K18ac peaks sensitive to β -oxidation inhibition. GO analysis reveals an enrichment for genes with roles in wing disc development.

While this observation is interesting, the authors have not shown a potential function for the higher levels of these two specific marks in the rim. The possible roles of these marks in regulating the expression of genes required in the development of the wing disc needs to be characterized more in detail. The paper relies heavily on imaging data and use of pharmacological inhibitors with specificity issues. All of these concerns preclude this study from being accepted for publication at this stage.

Specific comments:

The authors have used RNAi and overexpression constructs but none of those genetic manipulations (eg: Nejire and HDAC1 RNAi lines) were followed up by phenotypic analysis of the relevant areas in the emerging flies. This would be essential to show the functional importance of the observed H3K18ac and H4K8ac marks in the rim of the wing discs.

A study of this kind will benefit greatly from using one of the genetic platforms reported

(Günesdogan et al.2010, McKay et al. 2015, Graves et al.,2016, Zhang et al.,2018) to generate histone mutants that cannot be acetylated or mimic acetylation in position H3K18 or H4K8 (similar to Penke et al., Genetics 2018; Armstrong et al 2018; Zhang et al.,2018, Regadas I, Mol Cell 2021). In fact, Zhang et al., 2018, reports the generation of H4K8A and H3K18A and report no viability, fertility or other defects. This manuscript will benefit at many levels by the use of these or similar flies to address the specific role of the chromatin marks in question and their acetylation status.

Etomoxir has been shown to have several off-target effects, including inhibition of complex I of the electron transport chain (Raud et al.,Cell Metabolism 2018, Divakaruni et al., Cell Metabolism 2018, Connor et al., Scientific Reports 2018, Yao et al., Plos Biology 2018). Therefore orthogonal methods like genetic depletion of Cpt1 or other enzymes of the β -oxidation pathway are needed to support claims of higher β -oxidation in the rim as well as the gene expression differences upon β -oxidation inhibition.

The authors can stain for Cpt1 to check if its levels are higher in the rim vs the pouch.

For all the immunofluorescence images, please indicate how many discs were analyzed. Wherever a difference in the H3K18ac signal is seen, please quantify this difference in fluorescence signal (eg Fig 2a-b,e-f, ED Fig5). Analysis similar to the one in Fig 5f for the relative intensity of the histone marks studied as a function of the distance of the nuclei to the rim would be beneficial.

Please show representative images and describe in the text/figure legend, the changes in morphology reported in ED Fig 5.

The nuclear ACSS2 fluorescence intensity signal (Experiments shown in Fig 5e-h, ED Fig 8i, ED Fig 9) should be normalized with the signal from some other nuclear protein to account for staining variability.

The argument that the HAT activity of nejire correlates with that of its crotonylation activity is weak. It is true that the same catalytic center seems to be responsible for both HCT and HAT activity (Liu X et al., Cell discovery 2017). However, the two activities can be uncoupled and a mutation that reduces HAT while increases HCT has been reported in the paper.

The authors can also stain for the nejire and HDAC1 enzymes to check for differences in their protein levels in the rim vs rest of the wing disc tissue.

The levels of knockdown need to be reported wherever RNAi lines have been used. Please quantify the difference in the FISH signal upon nejire knockdown in Fig 3b.

The FISH experiments in Fig 6 are done upon nejire knockdown. The loss of Nejire affects several marks along with H3K18ac (H3K9ac, H3K27ac etc as seen in ED fig2a). Therefore the differences in expression of the assessed genes cannot be attributed directly to H3K18ac loss.

Author Rebuttals to Initial Comments:

Referee #1

This paper shows that specific histone acetylation marks (principally H3K9ac and H4K16ac) are selectively present in nuclei of cells comprising the rim of the *Drosophila* wing disc, a developmental model of a pseudo-stratified epithelium. Using a combination of genetic and pharmacological manipulations along with localization of specific histone acetylation marks, the authors build a case that selective acetylation of histones results from the specific access of nuclei in the rim to substrate Ac-CoA derived from mitochondrial produced metabolites. By perturbing proteins implicated in nuclear positioning, the LINC complex protein klaroid (*koi*) and the general myosin II contractility regulator Rho kinase, they argue that the selective labeling of histone in nuclei at the periphery depends on the position of the nucleus and not other factors.

Overall, the paper contains novel and interesting results on the selective marking of histones in nuclei of a pseudo-stratified epithelia, and intricate sleuthing to identify the proximal metabolic substrate that provides the source of Ac-CoA for the specific histone marks. Together these results point to the novel possibility that nuclear positioning may contribute to alterations in the epigenome. Yet, there are significant issues that need to be resolved before the paper is advanced to publication in a general interest journal such as *Nature*.

We thank the reviewer for the supportive comments and the thoughtful suggestions below, which we have tried to address as much as possible, and which we believe have led to a stronger manuscript.

I have three major concerns with the overall conclusion of the paper that the physical position of the nucleus determines its repertoire of histone marks and hence chromatin state:

1. First, the perturbations used to alter nuclear position (*lof* for *koi* and Rho kinase) both also generally affect mechanotransduction to the nucleus. Thus, as far as the authors have taken it, it is not possible to distinguish whether these *lof* perturbations are solely affecting nuclear position rather than other nuclear mechanical properties such as stiffness, chromatin compaction, or even nuclear/cytoplasmic transport, each of which may separately affect histone acetylation. Perhaps the authors will argue that probing such nuclear mechanical properties will entail studying mechanisms that they have not yet delved into, but without discerning amongst them, there is really no way to support the conclusion reflected in the title of the paper.

We thank the reviewer for this thoughtful comment. We now provide data showing that the effect of *koi* and *Rok* loss-of-function on H3K18ac is not due to other nuclear mechanical properties such as stiffness or chromatin compaction.

If the *Rok* and *koi* knockdowns were to cause increased H3K18ac due to a change in nuclear stiffness or chromatin compaction, then H3K18ac should increase in mutant nuclei regardless of nuclear location, since this would be a cell-intrinsic effect. We now provide data which show this is not the case. The *koi* and *Rok* knockdowns cause mutant nuclei to move basally in a passive, random fashion, as they get 'pushed around' by the other wildtype nuclei nearby that are actively moving apically. This means that many *koi* or *Rok* mutant nuclei are, by chance, still in the middle of the epithelium (see arrow in Reviewer Fig. 1):

We therefore quantified H3K18ac for a large number of wildtype and mutant nuclei, and plotted this against their distance from the tissue surface (Reviewer Fig. 2):

Reviewer Figure 2: Quantification of H3K18ac levels for wildtype (black) and *koi* mutant (green) nuclei plotted against their distance to the tissue surface. Nuclei that are located between the basal side of the disc and the middle of the disc were quantified.

This reveals that *koi* mutant nuclei that are in regions where wildtype nuclei are also present (from 8-25 µm from the surface) have H3K18ac levels that are indistinguishable from those of the wildtype nuclei. Hence, if the *koi* mutation is causing changes in chromatin compaction, nuclear stiffness or nuclear/cytoplasmic transport, this is not altering H3K18ac levels. Instead, only the nuclei that are very close to the tissue surface have high H3K18ac. In fact, one can see from this graph that the *koi* mutant nuclei are simply extending a position-dependent trend that is also visible in the wildtype nuclei. We now provide these data in Fig. 1n-o, and equivalent data for *Rok* in ED Fig. 1i-j.

2. Second, I am not convinced by the data presented that proximity of the nucleus to mitochondria does not contribute to the selective histone marks. The observation that mitochondria are present throughout the epithelia does not rule out the possibility that mitochondria are in much closer proximity to nuclei in the periphery. High magnification images and measurements of actual nuclear-mitochondrial distances are necessary.

We have now looked at this more extensively and provide in ED Fig. 8b-c a careful quantification of nuclear H3K18ac versus nuclear distance to the closest mitochondrion. We did this in clones expressing IFP to mark single cells, to make sure that we are measuring the distance between nuclei and mitochondria within one cell, and not mitochondria from a neighboring cell. This shows that H3K18ac^{high} nuclei are not closer to mitochondria than the other nuclei (coefficient of determination $r^2 = 0.05$).

3. Three, roughly half of the paper (Figures 3-5; ED Figures 6-8) comprises results identifying the Ac-CoA carbon source and enzymes. Yet, this aspect of the paper is not reflected in the title of the paper. This seems to be an oversight to me. If it turns out nuclear positioning is involved, isn't an equally important part of the story that there is localized generation of the substrate for the histone acetylases? To me the current title is not fully reflective of the results.

We agree with the reviewer and thank the reviewer for this comment. We have now changed the title to

"Nuclear position and localized acetyl-CoA production regulate chromatin state"

to fit within the limit of 75 characters (including spaces). If the Reviewer find this title too cryptic, we can go back to the original one, since the aspect of localized acetyl-CoA production is also discussed in the abstract.

Other comments:

4. Key to the authors' conclusion about nuclear positioning is their interpretation that in *koi* (or *Rjo* kinase) *lof* cells with nuclei displaced toward the periphery, the nuclei are positive for H3K18ac. They assume that the cells were negative for H3K18ac before *koi*/*Rho* kinase *lof*, yet it was unclear to me how they know this, particularly as there are many adjacent nuclei in non-*koi* expressing cells that are H3K18ac positive. Additionally, quantification is needed as only single examples are shown in Fig. 1k and ED Fig 1i.

We thank the reviewer for this comment, because it highlights the fact that we hadn't included quantifications of H3K18ac which clarify this point.

In the pouch of a wildtype wing disc, the most basal region is normally devoid of nuclei (Reviewer Fig. 3). The koi and Rok knockdowns cause some nuclei to move further basally than normal, into this region. Hence, we must assume that prior to formation of the clone, the nucleus was in this 'normal region' where the nuclei normally reside, and had H3K18ac levels that correspond to this region.

Reviewer Figure 3: In the pouch of the wing disc, the basal region is normally devoid of nuclei.

As the reviewer points out, some nuclei in the most basal part of this 'normal region' also have higher H3K18ac compared to the bulk (Reviewer Fig. 4). However, these levels are not as high as in the clone (arrowhead). Indeed, H3K18ac levels in the clone are higher than any other nucleus in the disc, where the nucleus resided prior to clone formation. Hence, we can conclude that H3K18ac levels increased.

Reviewer Figure 4: H3K18ac levels in a wing pouch containing a koi mutant clone (arrowhead).

Although we simplify the text by talking about "H3K18ac^{high}" and "H3K18ac^{low}", there is actually a continuum of H3K18ac levels. As discussed above in response to the Reviewer's first point, we now provide in Fig. 1n-o and ED Fig. 1i-j the quantifications of H3K18ac levels of wildtype and mutant nuclei as a function of distance to the tissue surface, which make this clear. It also shows that the koi mutant nuclei that are in this

most basal region (distance to the surface < 5µm) have higher H3K18ac levels than the wildtype nuclei (black dots). Hence, the knockdown caused the H3K18ac levels to increase. As the reviewer points out, there is a trend of increasing H3K18ac levels also in the wildtype cells as they get closer to the tissue surface, which further supports the concept of dependency of acetylation levels on the nuclear distance to the tissue surface.

5. In general, higher magnification images are needed to make the following results more convincing: a) the position of nuclei relative to the apical-basal aspect of the epithelium (Fig. 1g'; here it would also be useful to have apical and basal markers);

We now provide as ED Fig. 1f a higher magnification image of this disc (which is now Fig 1h-h'). We have marked in Fig. 1h' the apical and basal surfaces, (a molecular marker is not needed because we can determine this from the morphology), as well as in Figures 1i, 1n, ED1f, ED1i.

b) the proximity of nuclei to mitochondria (ED Fig. 7h-h'),

We now provide a quantification in ED Fig. 8c.

and c) whether the ACSS2 puncta are actually nuclear (Fig 5e-f, g-h).

We replaced Fig. 5e' and 5g' with higher magnification images. (Note that some ACCS2 puncta are nuclear, and some are cytosolic. ACCS2 is present in both compartments).

6. Some of the conclusions about loss/gain of signal are anecdotal and would benefit from more quantification (e.g., ED Fig 4a-b'''; Fig 3b, ED Fig 6b,g)

We thank the reviewer for prodding us into doing these quantifications, because it has improved the manuscript. We now provide a large number of new quantifications, all of which support the conclusions we originally drew in the original submission:

Fig 1o
Fig 2b, d, h, k
Fig 4e
ED Fig. 1j
ED Fig. 2d, f, i
ED Fig. 3b
ED Fig. 4b, d, f, h
ED Fig. 5h
ED Fig. 6c, f, l, n, p, r
ED Fig. 8c

ED Fig. 9e, g, i
ED Fig. 10g', h'

7. A caveat to the conclusion that high deacetylation does not occur in the interior is that there may be more than one deacetylase(s) (in addition to HDAC1) contributing to the signal in the interior.

We agree with the reviewer. To address this issue, we now provide new data showing that combined pharmacological inhibition of all deacetylases (both HDACs and sirtuins) does not increase pouch H3K18ac (ED Fig. 5g-h). This strengthens the conclusion that the H3K18ac rim is not due to de-acetylation in the interior.

8. The rescue by octanoate in Fig. 3g,h is partial and this should be indicated in the text. Similarly with HDAC1 kd in ED Fig. 6g.

Done - we have modified both the Results text and the figure legends.

Referee #2

This study by Willnow and Teleman shows a new link between fatty acid oxidation (FAO) and acetyl-CoA for histone acetylation in a spatially restricted manner. While the direct link between FAO and histone acetylation is known, this study shows that histone acetylation is required for proper gene expression and development in the outer rim disc, but not other parts of the *Drosophila* wing disc. Generally, this study is well-controlled and provides a logical flow between hypothesis and experiment.

We thank the reviewer for the supportive comments.

I can only offer one major critique:

MAJOR:

The study's biggest limitation is that the authors don't know how proximity to the tissue surface influences nuclear ACSS2 localization and generation of substrate for NEJ. They convincingly show that fatty acid β -oxidation is high in the rim of the disc, and that this region also has more (generally?) nuclear ACSS2 localization. Both of these events are required

for histone acetylation and gene expression, but the authors don't know *how* this occurs. The key idea and novelty of the manuscript are in how the physical position of a nucleus within a tissue affects its chromatin state. But as presented, the authors don't know the mechanism(s) explaining how positioning influences metabolism, chromatin, and gene expression.

We agree with the reviewer. We have been working hard for the past year to address this issue, but it appears to be a very hard nut to crack.

The model that is most consistent at this moment with all the data presented in the manuscript is that there is a signal coming from the hemolymph that generates an intracellular gradient of either a signaling protein or a metabolite that is high near the tissue surface, and decreases further in. For instance, it could be a kinase such as PDK which is anchored at the cell surface, or a metabolite such as calcium or cAMP which is generated in a localized fashion. When nuclei enter this region of high signal, something happens to them (e.g. phosphorylation of nuclear pores?) that causes ACSS2 to enter and accumulate. We do not think ACSS2 is modified by this signal, because otherwise ACSS2 would be able to enter the nucleus no matter where the nucleus is in the cell, since ACSS2 is small enough to move around (just like transcription factors that are regulated by phosphorylation and can enter the nucleus no matter where it is.)

We have done many experiments to add mechanistic details to this model. We can categorize most of our results thus far into two groups: 1) manipulations that don't affect H3K18ac, and 2) manipulations that cause a drop in H3K18ac but also cause the wing disc tissue to look stressed. Interpretation of the results in this second category will require additional work, to distinguish whether each manipulation is reducing cell viability and as a consequence leading to loss of H3K18ac (i.e. not interesting), or whether the pathway we are manipulating affects both H3K18ac and viability independently of each other. Some examples are listed below:

We have tested a possible role for signaling pathways that affect tissue growth, such as:

- AGC kinases, by either overexpressing them, knocking them down, or treating discs with pharmacological activators or inhibitors. We have done RNAi for PKA, PKG isoforms, PKC, and S6K, and we have overexpressed constitutively active and dominant negative PKA. We have treated discs with H89, H8, and H7 to inhibit PKA, PKG, and PKC and we have activated PKA with forskolin or cell permeable cAMP, or PKG with cell permeable cGMP. None of these had an effect on H3K18ac.

- The PI3K pathway, by overexpressing wildtype or constitutively active PI3K (Dp110), by knocking down Akt, Rictor or S6K, by overexpressing PTEN, or by pharmacologically inhibiting Akt with MK2206. Again, no effect on H3K18ac.
- The Hippo pathway (which also senses physical forces), by knocking down warts or yorkie (the fly ortholog of YAP) as well as by overexpressing yorkie, but none had an effect on H3K18ac.
- The Ras/MAPK pathway by overexpressing Ras-V12 (no effect on H3K18ac).
- Regulators of cellular contractility, for instance knocking down Rho kinase. No effect.

We have also aimed to manipulate some key intracellular signaling metabolites by treating discs with:

- butyryl-cAMP (mentioned above) - no effect
- SQ22536 (adenylate cyclase inhibitor) - no effect
- cGMP (mentioned above) - no effect
- ionomycin to increase cytosolic calcium levels. This caused a drop in H3K18ac but also obvious toxicity.
- ST-18-COH3, a phospho-lipase C inhibitor to reduce DAG levels - no effect at 10 μ M and at 50 μ M it causes a drop in both H3K18ac and cell viability.

The most intriguing result we obtained so far is that treatment of discs with the drug thioridazine causes H3K18ac to drop. (This was by chance - we tested thioridazine because it was reported to inhibit peroxisomal beta-oxidation, and we wanted to test if peroxisomal beta-oxidation contributes to this system. We also tested this genetically using pex19 null mutants and found no effect, excluding a peroxisomal cause of the phenotype.) Treatment with thioridazine for 30 min causes a mild decrease in H3K18ac levels and the discs look otherwise fine. The drop is stronger at 120 min (complete loss of H3K18ac) but then the discs also looked stressed. Nonetheless, a drop at 30 minutes without morphological defects suggests this might be a specific effect.

Thioridazine inhibits a broad set of GPCRs such as serotonin receptor, alpha-adrenergic receptor, beta-adrenergic receptor, dopamine receptor and acetylcholine receptor, as well as hERG, a potassium channel. Intriguingly, wing discs express a number of neurotransmitter receptors such as the dopamine receptor orthologs Dop1R2, CG12290 and DopEcR, albeit at low levels. Since it is possible that a signal is coming from the hemolymph, we sought to dissect the activity of thioridazine by testing more specific inhibitors. We tested SKF83566 and eticlopride (dopamine receptor antagonists) and imidacloprid (inhibiting nicotinic acetylcholine receptors), however none of them phenocopied thioridazine. We also tried removing potassium from the explant medium, and this efficiently abrogated H3K18ac in the wing disc. However, the wing discs also appear stressed, complicating the interpretation. It is possible that potassium is both needed for cell viability and that it forms an intracellular

gradient for signaling purposes. As the reviewer can appreciate, these results are intriguing, but too preliminary to present.

In conclusion, we think it will require an entire additional study and several years to understand this mechanism. Nonetheless, we believe the current study already provides a large amount of mechanistic insight, identifying nuclear position as a regulatory input, and narrowing down the mechanism to something around ACSS2.

MINOR:

Beyond inhibiting CPT1, Etomoxir at high concentrations consumes free CoA and disrupts many metabolic pathways. Genetic ablation of CPT1 would further strengthen the evidence for FAO.

We agree fully with the reviewer. Although this seems like a simple request, it is a complex issue which we have struggled with, and would like to reply by addressing three questions:

- 1) Does genetic ablation of CPT1 phenocopy etomoxir? (no)
- 2) Is the source of acetyl-CoA FAO? (yes)
- 3) Is etomoxir working on-target? (probably yes)

1) does genetic ablation of CPT1 phenocopy etomoxir?

We did indeed try genetic ablation of the fly CTP1 ortholog, withered (whd): We obtained a whd RNAi line from the VDRC stock center, and we generated three new transgenic RNAi fly lines ourselves, which we validated to be 75-90% effective by Q-RT-PCR. None of them affected H3K18ac. We obtained a whd knockout line from the Bloomington public repository, but this line did not have the characteristic withered wing phenotype, indicating it had either lost the mutation or gained suppressor mutations over time. We also obtained a CPT1 knockout allele, kindly provided by Alistair Coulthard (PMID 18521119) which had withered wings, but it also had no H3K18ac phenotype. We think this combined dataset, and in particular the null allele from Alistair Coulthard, proves that ablation of whd alone does not phenocopy the etomoxir effect. In case etomoxir might also be inhibiting medium and short-chain transporters, we tried combined quadruple knockdown of whd, the two short-chain transporters (CRAT and CG5265) and the medium-chain transporter crot, but this also did not affect H3K18ac. For this last experiment, however, the lack of a phenotype might be due to poor RNAi-efficiency when multiple different RNAi constructs are combined (due to dilution of the GAL4).

The problem is that there is likely a functionally-redundant long-chain fatty acid transporter in Drosophila, just like there are 3 CPT1 isoforms in mice. Indeed, in mice, loss of one of these three isoforms, CPT-1b, is lethal (PMID 18023382), whereas loss of whd in Drosophila is not - the flies are viable with only mild morphological defects

and hypersensitivity to oxidative stress (PMID 18521119) - suggesting the presence of a functionally redundant transporter. This other redundant transporter, however, has not yet been reported. We reviewed a manuscript in 2020 which 1) showed that *Drosophila* efficiently catabolizes long chain fatty acids in the absence of whd, and 2) identified a mitochondrially-localized transporter that abolished long-chain fatty acid beta-oxidation when mutated in combination with whd loss-of-function. Although we gave that manuscript a positive, supportive review, it has not yet been published so we cannot yet use this information.

2) Is the source of acetyl-CoA FAO?

All that said, our conclusion that FAO is the source of the acetyl-CoA does not come solely from the etomoxir treatment, but from multiple datasets combined:

- Since we were also aware that etomoxir can have off-target effects, we rescued the etomoxir effect by treating discs with octanoate.
- One could argue that if etomoxir is generally reducing CoA levels or dampening respiration (Complex I activity), then providing more substrate in any form (octanoate, glucose or glutamine) would rescue the defect. However, the explant medium we are using (Schneider's) contains 10x more glucose and 10x more glutamine than larval hemolymph, yet etomoxir reduces H3K18 acetylation in this medium. This means that the etomoxir effect can be rescued by octanoate, but not by glucose or by glutamine, indicating it is specific for lipids.
- We found that inhibition of the other main sources of acetyl-CoA - glycolysis and glutaminolysis - do not affect the H3K18ac rim (Fig. 3e-f, ED Fig. 6 g-j). We did this both pharmacologically and genetically. So, basically, this only leaves fatty acid beta-oxidation as the possible source.
- The H3K18ac rim and the high mitochondrial membrane potential anti-correlate with BODIPY staining of neutral-lipids (Fig. 3k), indicating there is indeed high fatty acid catabolism in this region.
- Finally, we see the same phenotypes when looking at mitochondrial membrane potential (ED Fig. 7l-m) - inhibition of glycolysis or glutaminolysis does not remove the high membrane potential in the rim, but etomoxir does.

So considering all these data together, we think we can safely conclude that FAO is the source of the acetyl-CoA, which is the main conclusion we want to draw.

3) Is etomoxir working on-target?

In addition to the fact that the etomoxir effect can be rescued by octanoate but not glucose or glutamine, we now also provide two additional datasets to look at this:

- We performed a titration of etomoxir to lower concentrations (ED Fig. 6k-l and Reviewer Figure 5 below) and find that it already reduces H3K18ac at 50 μ M. This supports an on-target effect given that off-target effects are not seen below 100 μ M (PMID 30043752) and that we usually need to use drugs at higher concentrations on *Drosophila* tissues compared to human cell culture.

Reviewer Figure 5: Titration of etomoxir and effect on wing disc H3K18ac.

- To exclude that etomoxir is acting on H3K18ac via inhibition of Complex I, we treated wing discs with the potent Complex I inhibitor rotenone (ED Fig. 6m-n). This shows that rotenone does not cause a drop in H3K18ac that is as strong as etomoxir. This is not due to inefficient inhibition of Complex I by rotenone because higher doses of rotenone do not cause a stronger drop in H3K18ac (ED Fig. 6o-p). (This makes sense because FABO fuels respiration via both complex I and complex III, and in fact the acetyl-CoA used for acetylation is the acetyl-CoA that is not respired.)

Altogether, we think the octanoate rescue plus these two additional pieces of data support the idea that etomoxir is likely acting on-target. Nonetheless, as mentioned above, the important conclusion that FABO is the source of the acetyl-CoA comes from also the other sets of data.

In sum, we are sorry that although the reviewer is generally positive, we are not able to fully address the two issues raised by the reviewer. Indeed, the reviewer has put his/her finger on exactly the two issues which we are still working on for the future. We hope nonetheless the reviewer finds the manuscript interesting enough.

Referee #3

In the manuscript, Willnow et al. report higher levels of total lysine acetylation, H3K18ac and H4K8ac marks in the outer rim of the *Drosophila* wing discs. H3K9ac, H3K27ac and some methylation marks (H3K9me1/2/3, H3K27me3) studied do not show the same enrichment. Nejire is identified as the main acetyltransferase responsible for the deposition of the H3K18ac mark. However, nej mRNA levels are uniform in the wing disc. Unlike inhibition of glycolysis and glutaminolysis, inhibition of fatty acid β -oxidation via etomoxir treatment decreases the H3K18ac levels in the rim. This suggests that β -oxidation is the main source of acetyl-CoA in the rim. The authors claim there is higher β -oxidation in the rim based on the higher mitochondrial membrane potential observed, which is sensitive to etomoxir treatment. In addition, higher nuclear levels of the enzyme ACSS2 are reported in the wing disc rim. The authors suggest that the conversion of acetate to acetyl-CoA by the nuclear ACSS2 population is the rate limiting step in moderating H3K18ac levels in the wing disc rim. Finally with a H3K18ac Cut & Run experiment with and without the treatment of the wing discs with etomoxir, the authors identify the H3K18ac peaks sensitive to β -oxidation inhibition. GO analysis reveals an enrichment for genes with roles in wing disc development.

While this observation is interesting, the authors have not shown a potential function for the higher levels of these two specific marks in the rim. The possible roles of these marks in regulating the expression of genes required in the development of the wing disc needs to be characterized more in detail. The paper relies heavily on imaging data and use of pharmacological inhibitors with specificity issues. All of these concerns preclude this study from being accepted for publication at this stage.

We thank the reviewer for the careful reading of our manuscript and for the helpful suggestions. Replies to the specific comments are below. We hope to have addressed all the issues raised, and thank the reviewer for having contributed towards making the manuscript stronger.

Specific comments:

The authors have used RNAi and overexpression constructs but none of those genetic manipulations (eg: Nejire and HDAC1 RNAi lines) were followed up by phenotypic analysis of the relevant areas in the emerging flies. This would be essential to show the functional importance of the observed H3K18ac and H4K8ac marks in the rim of the wing discs.

We now provide several sets of data showing that the observed histone marks in the rim are indeed important for fly development.

First, it is important to point out that inhibition of beta-oxidation leads to a strong drop in total acetyl-lysine (ED Fig. 7c-c',f), and not just H3K18ac and H4K8ac. We only use H3K18ac and H4K8ac as representative marks. However, the mechanism we describe affects acetate and acetyl-CoA levels and hence bulk histone acetylation, and therefore one needs to look at the functional importance of all of these marks combined. As the reviewer points out, this can be done with nej knockdown.

- A strong knockdown of either nej or HDAC1 leads to lethality of cells and animals. However, this is not a 'fair' analysis because these manipulations reduce histone acetylation more strongly than the drop caused by impaired fatty acid beta-oxidation. We therefore didn't put these data in the manuscript.

- We next titrated down the strength of the knockdowns by using weaker drivers and lower temperatures. Knockdown of nej in the rim with tshG4 causes pupal lethality at all temperatures, even the lowest possible 18°C. This per se already shows the importance of these acetylation marks in the rim.

- Instead, we found that enG4 (which is a very weak driver) at 18°C gave viable animals for both nej RNAi lines used in this study and at least one of the HDAC1 RNAi lines. Under these conditions, the drop in H3K18ac is very mild (see ED Fig. 10g-h', comparing anterior to posterior compartments) - indeed it is weaker than the effect caused by etomoxir. Despite the mild drop in H3K18ac, this condition leads to strong developmental defects with both nej and HDAC1 knockdowns (Fig. 6i-l').

- Finally, an intriguing piece of data is that knockdown of nej in the rim with tsh-GAL4 causes strong and rapid loss of the tissue (the GFP-expressing rim is gone), whereas knockdown of nej in the pouch with nubbin-GAL4 results in a much weaker phenotype - the pouch is mildly smaller, but GFP domain is still present (Reviewer Fig. 6) - indicating that histone acetylation is more essential in the rim than in the pouch.

Worth mentioning is that in the original submission we looked at the consequences of reduced histone acetylation on gene expression, specifically in the rim of the disc. Here we found that expression of important developmental genes such as *mirror* and *homothorax* is affected (Fig. 6 e-h', ED Fig. 10 b-f'). This is also a phenotype and shows functional importance.

A study of this kind will benefit greatly from using one of the genetic platforms reported (Günesdogan et al. 2010, McKay et al. 2015, Graves et al., 2016, Zhang et al., 2018) to generate histone mutants that cannot be acetylated or mimic acetylation in position H3K18 or H4K8 (similar to Penke et al., Genetics 2018; Armstrong et al 2018; Zhang et al., 2018, Regadas I, Mol Cell 2021). In fact, Zhang et al., 2018, reports the generation of H4K8A and H3K18A and report no viability, fertility or other defects. This manuscript will benefit at many levels by the use of these or similar flies to address the specific role of the chromatin marks in question and their acetylation status.

As suggested by the reviewer, we now present data using the system reported by Alf Herzig and colleagues (Günesdogan et al. 2010).

Since the mechanism we describe here affects both H3K18ac and H4K8ac, as well as other marks, we decided to analyze mutant flies in which both H3K18 and H4K8 are simultaneously mutated. Furthermore, since Zhang et al. mutated the lysines to alanine, which causes a loss of the positive charge of lysine, thereby mimicking the effect of acetylation (rather than mimicking loss of acetylation), we aimed to mutate the lysines to arginine, which retains positive charge but cannot be acetylated. Thus, this required the generation of new transgenic lines.

To this end, since the Reviewer specifically mentions Zhang et al., we first tried contacting the corresponding authors of Zhang et al. to ask for the necessary fly lines and plasmids. We emailed Guanjun Gao twice at gaogj@shanghaitech.edu.cn (which is still his current affiliation) but he never replied. So we also emailed Junbiao Dai at junbiao.dai@siat.ac.cn and also got no reply. Hence, we were unable to test if the lack of a phenotype in their single H3K18 mutant flies is due to compensatory increases in acetylation of other sites such as H4K8 in the rim of the wing disc.

We were thankful that instead Alf Herzig was very responsive and helpful and shared his reagents with us. We thereby generated flies carrying mutations in the endogenous histone locus and carrying 12 copies of a histone complex transgene where both H3K18 and H4K8 are mutated to either the non-acetyltable arginine or to glutamine to mimic the loss of positive charge caused by acetylation. (Alf Herzig mentioned that in doing so, one needs to be careful that the plasmids do not recombine in the bacteria to remove one or two copies of the tandem histone loci, and that the transgenic flies get the expected full complement of histone copies. We confirmed this by quantitative PCR on genomic DNA from single fly embryos to confirm that they have 12 copies of the histone locus). The end-result is that both K->R and K->Q lines are embryonic or very early larval lethal. Therefore, unlike the single mutants mentioned by the reviewer, double mutation of both H3K18 and H4K8 is lethal, highlighting the functional importance of these marks.

Etomoxir has been shown to have several off-target effects, including inhibition of complex I of the electron transport chain (Raud et al., Cell Metabolism 2018, Divakaruni et al., Cell Metabolism 2018, Connor et al., Scientific Reports 2018, Yao et al., Plos Biology 2018). Therefore orthogonal methods like genetic depletion of Cpt1 or other enzymes of the β -oxidation pathway are needed to support claims of higher β -oxidation in the rim as well as the gene expression differences upon β -oxidation inhibition.

We agree fully with the reviewer. Although this seems like a simple request, it is a complex issue which we have struggled with, and would like to reply by addressing three questions:

- 1) Does genetic ablation of CPT1 phenocopy etomoxir? (no)
- 2) Is the source of acetyl-CoA FAO? (yes)
- 3) Is etomoxir working on-target? (probably yes)

1) does genetic ablation of CPT1 phenocopy etomoxir?

We did indeed try genetic ablation of the fly CTP1 ortholog, withered (whd): We obtained a whd RNAi line from the VDRC stock center, and we generated three new transgenic RNAi fly lines ourselves, which we validated to be 75-90% effective by Q-RT-PCR. None of them affected H3K18ac. We obtained a whd knockout line from the Bloomington public repository, but this line did not have the characteristic withered wing phenotype, indicating it had either lost the mutation or gained suppressor mutations over time. We also obtained a CPT1 knockout allele, kindly provided by Alistair Coulthard (PMID 18521119) which had withered wings, but it also had no H3K18ac phenotype. We think this combined dataset, and in particular the null allele from Alistair Coulthard, proves that ablation of whd alone does not phenocopy the etomoxir effect. In case etomoxir might also be inhibiting medium and short-chain transporters, we tried combined quadruple knockdown of whd, the two short-chain transporters (CRAT and CG5265) and the medium-chain transporter crot, but this also did not affect H3K18ac. For this last experiment, however, the lack of a phenotype might be due to poor RNAi-efficiency when multiple different RNAi constructs are combined (due to dilution of the GAL4).

The problem is that there is likely a functionally-redundant long-chain fatty acid transporter in *Drosophila*, just like there are 3 CPT1 isoforms in mice. Indeed, in mice loss of one of these three isoforms, CPT-1b, is lethal (PMID 18023382), whereas loss of whd in *Drosophila* is not - the flies are viable with only mild morphological defects and hypersensitivity to oxidative stress (PMID 18521119) - suggesting the presence of a redundant transporter. This other redundant transporter, however, has not yet been reported. We reviewed a manuscript in 2020 which 1) showed that *Drosophila* efficiently catabolizes long chain fatty acids in the absence of whd, and 2) identified a mitochondrially-localized transporter that abolished long-chain fatty acid beta-oxidation when mutated in combination with whd loss-of-function. Although we gave that manuscript a positive, supportive review, it has not yet been published so we cannot yet use this information.

Likewise, the other steps of beta-oxidation also have redundant enzymes. We tried double knockdowns of the *Drosophila* thiolases (yip2 plus either of the two subunits of the mitochondrial trifunctional protein mtpa or thiolase) but we were not able to block beta-oxidation, likely because a double knockdowns are not very efficient due to dilution of the GAL4. Generally, we have found it difficult to block also other central metabolic pathways with knockdowns (such as glycolysis or respiration), likely because metabolic enzymes are amongst the most highly expressed genes, and they act catalytically. In sum, despite extensive efforts, we were not able to find a genetic condition that efficiently inhibits beta-oxidation.

2) Is the source of acetyl-CoA FAO?

All that said, our conclusion that FAO is the source of the acetyl-CoA does not come solely from the etomoxir treatment, but from multiple datasets combined:

- Since we were also aware that etomoxir can have off-target effects, we rescued the etomoxir effect by treating discs with octanoate.
- One could argue that if etomoxir is generally reducing CoA levels or dampening respiration (Complex I activity), then providing more substrate in any form (octanoate, glucose or glutamine) would rescue the defect. However, the explant medium we are using (Schneider's) contains 10x more glucose and 10x more glutamine than larval hemolymph, yet etomoxir reduces H3K18 acetylation in this medium. This means that the etomoxir effect can be rescued by octanoate, but not by glucose or by glutamine, indicating it is specific for lipids.
- We found that inhibition of the other main sources of acetyl-CoA - glycolysis and glutaminolysis - do not affect the H3K18ac rim (Fig. 3e-f, ED Fig. 6 g-j). We did this both pharmacologically and genetically. So, basically, this only leaves fatty acid beta-oxidation as the possible source.
- The H3K18ac rim and the high mitochondrial membrane potential anti-correlate with BODIPY staining of neutral-lipids (Fig. 3k), indicating there is indeed high fatty acid catabolism in this region.
- Finally, we see the same phenotypes when looking at mitochondrial membrane potential (ED Fig. 7l-m) - inhibition of glycolysis or glutaminolysis does not remove the high membrane potential in the rim, but etomoxir does.

So considering all these data together, we think we can safely conclude that FAO is the source of the acetyl-CoA, which is the main conclusion we want to draw.

3) Is etomoxir working on-target?

In addition to the fact that the etomoxir effect can be rescued by octanoate but not glucose or glutamine, we now also provide two additional datasets to look at this:

- We performed a titration of etomoxir to lower concentrations (ED Fig. 6k-l and Reviewer Figure 7 below) and find that it already reduces H3K18ac at 50 μ M. This supports an on-target effect given that off-target effects are not seen below 100 μ M (PMID 30043752) and that we usually need to use drugs at higher concentrations on *Drosophila* tissues compared to human cell culture.

Reviewer Figure 7: Titration of etomoxir and effect on wing disc H3K18ac.

- To exclude that etomoxir is acting on H3K18ac via inhibition of Complex I, we treated wing discs with the potent Complex I inhibitor rotenone (ED Fig. 6m-n). This shows that rotenone does not cause a drop in H3K18ac that is as strong as etomoxir. This is not due to inefficient inhibition of Complex I by rotenone because higher doses of rotenone do not cause a stronger drop in H3K18ac (ED Fig. 6o-p). (This makes sense because FABO fuels respiration via both complex I and complex III, and in fact the acetyl-CoA used for acetylation is the acetyl-CoA that is not respired.)

Altogether, we think the octanoate rescue plus these two additional pieces of data support the idea that the etomoxir is likely acting on-target. Nonetheless, as mentioned above, the important conclusion that FABO is the source of the acetyl-CoA comes from also the other sets of data.

The authors can stain for Cpt1 to check if its levels are higher in the rim vs the pouch.

No antibody detecting *Drosophila* Cpt1 has been reported. Therefore, we screened commercially available antibodies that detect human Cpt1 to look for those that were generated using epitopes that are conserved between humans and flies, and tested two of them, Proteintech 15184-1-AP and Sigma SAB5700800. Both, however, gave weak and diffuse, cytosolic signals (whereas CPT1 is mitochondrially localized). We also tested their specificity by staining wing discs with a whd/Cpt1 knockdown in the posterior half of the wing (using an RNAi line from the VDRC and an RNAi line which we generated ourselves and confirmed by Q-RT-PCR to be 90% efficient at the RNA level), and neither antibody gave reduced signal in the posterior, knockdown compartment. Therefore, unfortunately we do not have an antibody to detect fly CPT1.

We assume the reviewer is asking whether CPT1 is enriched in the rim because this would explain the increased beta-oxidation in that domain (good thought!). Since metabolic enzymes are often regulated at the mRNA level, we checked some data

which we previously published where we generated expression maps of all genes in the wing disc by dissociating discs into single cells, sequencing them (scRNA-seq), and then bioinformatically recompiling these data to generate spatial expression maps (PMID 31363221). Unfortunately the map for whd does not show enrichment in the rim. So at least for now, we have no data suggesting that the elevated beta-oxidation in the rim is due to elevated CPT1 protein levels there.

For all the immunofluorescence images, please indicate how many discs were analyzed.

We have added samples sizes to all the figure legends.

Wherever a difference in the H3K18ac signal is seen, please quantify this difference in fluorescence signal (eg Fig 2a-b,e-f, ED Fig5). Analysis similar to the one in Fig 5f for the relative intensity of the histone marks studied as a function of the distance of the nuclei to the rim would be beneficial.

We added quantifications (including H3K18ac levels vs distance of nuclei to the surface) as

Fig 1o

Fig 2b, d, h, k

Fig 4e

ED Fig. 1j

ED Fig. 2d, f, i

ED Fig. 3b

ED Fig. 4b, d, f, h

ED Fig. 5h

ED Fig. 6c, f, l, n, p, r

ED Fig. 8c

ED Fig. 9e, g, i

ED Fig. 10g', h'

Please show representative images and describe in the text/figure legend, the changes in morphology reported in ED Fig 5.

To be able to show these images, we have now shifted the table that was ED Fig. 5 into Supplemental Table 1, and now provide all the disc images as ED Figs. 2-5.

The nuclear ACSS2 fluorescence intensity signal (Experiments shown in Fig 5e-h, ED Fig 8i, ED Fig 9) should be normalized with the signal from some other nuclear protein to account for staining variability.

The reviewer raises an important point: that signal intensity may possibly drop as one images deeper into the tissue due to optical and permeability reasons. This is true for both our ACSS2 quantifications and our H3K18ac quantifications. As suggested, to control for this we now provide as ED Fig. 9c-e the staining for another nuclear protein, H3, and find that this does not show any trend as a function of distance ($r^2 = 0.03$).

The argument that the HAT activity of nejire correlates with that of its crotonylation activity is weak. It is true that the same catalytic center seems to be responsible for both HCT and HAT activity (Liu X et al., Cell discovery 2017). However, the two activities can be uncoupled and a mutation that reduces HAT while increases HCT has been reported in the paper.

We thank the reviewer for pointing that out. We can think of two possible mechanistic interpretations of the data shown in Liu X et al., one relevant for our manuscript and one not:

1) One option is that the point mutation alters the structure or activity of CBP/p300 in such a way that its intrinsic catalytic activity (e.g. in vitro) for acetylation is reduced while its intrinsic catalytic activity for crotonylation is not. For instance, a mutation that impairs binding of CBP to acetyl-CoA but not to crotonyl-CoA would be an example. In this case, these point mutations are not relevant for our manuscript because nej protein is not differentially mutated in the rim versus the pouch, and therefore its intrinsic HCT vs HAT activity cannot be altered in the two domains.

2) The other option is that these mutations alter the interaction of CBP with proteins that differentially help it crotonylate versus acetylate. In this case, it is entirely possible that nej interacts with different proteins in the rim than in the pouch, and therefore its HAT activity increases there, but its HCT activity does not. Indeed, this could be part of the mechanism how nuclear position regulates acetylation in the rim. We have added a sentence to this effect in the manuscript.

(That said, we see that H3K9ac also goes up specifically in the rim, and not in the pouch, when we treat discs with acetate. Since H3K9ac in the wing disc is GCN5 dependent, and not nej-dependent, this indicates the mechanism is not specific to nej. Thus our data fits more simply with the interpretation that nuclear acetyl-CoA availability is altered in the rim, rather than both nej and GCN5 changing their interactome.)

The authors can also stain for the nejire and HDAC1 enzymes to check for differences in their protein levels in the rim vs rest of the wing disc tissue.

We now provide data showing that:

- nej protein is uniform in the disc in Figure 3a (with a knockdown control for antibody specificity in Fig. 3b and ED Fig. 5l).
- HDAC1 protein is uniform in ED Fig. 5i with knockdown controls in ED Fig. 5j-k.

The levels of knockdown need to be reported wherever RNAi lines have been used. Please quantify the difference in the FISH signal upon nejire knockdown in Fig 3b.

We now provide quantifications or validations for knockdowns used in the manuscript (including the FISH signal, now ED Fig. 6) either at the mRNA level or by looking at the target protein (e.g. nej, HDAC1, PDH, etc).

(For the screen where we tested many different acetyltransferases and deacetylases - now Suppl. Table 1 - we did not test the knockdown efficiency of each line, but instead used multiple different lines for each gene. We're providing these data because they might be useful for the community, although they are not really necessary in this manuscript. If necessary, we can remove them.)

The FISH experiments in Fig 6 are done upon nejire knockdown. The loss of Nejire affects several marks along with H3K18ac (H3K9ac, H3K27ac etc as seen in ED fig2a). Therefore the differences in expression of the assessed genes cannot be attributed directly to H3K18ac loss.

We thank the reviewer for this comment, because it made us realize that we had incorrectly phrased this paragraph in the Results to focus on H3K18ac. However inhibition of beta-oxidation does not only cause a drop in H3K18ac, but also in H4K8, H3K9, H3K27 and indeed total acetylated lysine (ED Fig. 7a-f), and we use H3K18ac throughout the manuscript only as one exemplary acetylation. Our purpose in Fig 6 is not to test the consequence of H3K18ac, but the consequence of the generalized increase in histone acetylation in the rim, seen also with total acetyl-lysine antibody (Fig. 1e). We have now reworded the text to broaden it to histone acetylation more generally.

Reviewer Reports on the First Revision:

Referees' comments:

Referee #1 (Remarks to the Author):

The authors have done a nice job addressing my comments (and those of the other reviewers) and the manuscript is much improved. In particular, they have satisfactorily addressed my three major concerns with the original manuscript. These include: showing that other aspects of nuclear mechanotransduction do not contribute to the selective histone acetylation; including clearer data that proximity of mitochondria to nuclei is not contributing; and modifying the title to include the idea of localized Ac-CoA production. Quantification of most results has also been included as requested. In my opinion the authors have also done an exemplary job of addressing the key concerns of the other reviews.

There are relatively few reports in the scientific literature that establish such a clear mechanistic advance in how nuclear positioning influences cell behavior. This case is particularly notable as it connects metabolism with epigenetic events and downstream *Drosophila* development. Although there are mechanistic details yet to be learned (particularly those raised by Reviewer 2), I am strongly in favor of accepting the paper for publication in *Nature*.

Referee #2 (Remarks to the Author):

I have no further comments.

Referee #3 (Remarks to the Author):

The revised manuscript has improved upon revision. Although, the authors were not able to address some of the concerns raised, they provided reasonable explanations where necessary.

Overall, I do find the distinct localisation pattern of histone acetylation in Wing Discs interesting, so although the mechanism of how exactly this pattern is achieved is missing, I support publication of this study.